# SpiR is a gut microbial enzyme that drives cholesterol conversion

Gabriela Arp [1,5], Sophia Levy [1,5], Angela K. Jiang [1,2],
Keith Dufault-Thompson[2], Aoshu Zhong [3], Maggie Grant [1], Yue Li[4],
Xiaofang Jiang [2] ✉ & Brantley Hall[1,4] ✉

The gut microbiota contributes to cholesterol homeostasis by converting cholesterol into coprostanol, a non-absorbable sterol excreted in the feces. However, the enzymes mediating this process remain poorly defined. Here, we identify *spiR*, a steroid $\Delta^{5\text{-}4}$ isomerase/3-keto reductase from *Eubacterium coprostanoligenes* that catalyzes the initial oxidation of cholesterol to cholestenone, a requisite step in coprostanol production. We confirm that SpiR oxidizes both cholesterol and pregnenolone, and stereospecifically reduces 3-keto-steroids to 3β-hydroxylated forms. We show that SpiR preferentially binds to cholesterol over related steroids and functions as an NAD(H)-dependent homodimer. Through phylogenetic analysis, we show that *spiR* clusters with known $\Delta^{5\text{-}4}$ isomerases and is restricted to an uncultured clade within Acutalibacteraceae, where it frequently co-occurs with species encoding *ismA*, a gene previously implicated in cholesterol conversion. We analyze a multi-omic dataset from three human cohorts and find that *spiR* homologs were strongly enriched in individuals exhibiting cholesterol conversion. We also show that *spiR* homologs have a greater predictive power for cholesterol conversion than *ismA* homologs, establishing them as superior markers of microbial cholesterol metabolism. Our findings refine the enzymatic model of cholesterol metabolism in the gut and establish *spiR* as a critical biomarker and mechanistic driver for microbiome-mediated cholesterol reduction.

Hypercholesterolemia is a major risk factor for cardiovascular disease, the leading cause of global mortality. While diet and genetics are major contributors to elevated serum cholesterol levels, the gut microbiome is being increasingly recognized as a key aspect of cholesterol homeostasis. Gut bacteria affect cholesterol metabolism by converting cholesterol into coprostanol, a non-absorbable sterol that can be excreted as a waste product, thereby reducing serum cholesterol[1]. Approximately one gram of cholesterol enters the intestine daily from a combination of diet, bile, and epithelial turnover; 50–60% is reabsorbed, while the rest is excreted or metabolized by the gut microbiota[2,3].

An inverse correlation between fecal coprostanol-to-cholesterol ratios and serum cholesterol levels has been reported, suggesting that individuals with efficient gut microbial cholesterol conversion tend to have lower circulating cholesterol[4]. Early evidence of a cholesterol-lowering effect associated with microbial cholesterol metabolism was found in the 1970s[5]. Studies show that individuals classified as low converters of cholesterol to coprostanol often have higher serum cholesterol levels than those classified as high converters[4,6,7]. The microbial origin of this process is reinforced by the finding that germ-free animals do not convert cholesterol to coprostanol and exhibit

[1]Department of Cell Biology and Molecular Genetics, University of Maryland, College Park, College Park, MD, USA. [2]National Library of Medicine, National Institutes of Health, Bethesda, MD, USA. [3]Department of Biology, Wake Forest University, Winston-Salem, North Carolina, USA. [4]Center for Bioinformatics and Computational Biology, University of Maryland, College Park, College Park, MD, USA. [5]These authors contributed equally: Gabriela Arp, Sophia Levy. ✉e-mail: xiaofang.jiang@nih.gov; brantley@umd.edu

higher circulating cholesterol levels[5]. Similarly, antibiotic treatment has been shown to reduce cholesterol-to-coprostanol conversion and correspondingly increase serum cholesterol levels[5,8]. To further support microbial involvement, a study found the abundance of coprostanol-producing bacteria in the gut is associated with greater cholesterol conversion efficiency and lower serum cholesterol levels[9]. These findings underscore the potential of microbial cholesterol metabolism to act as a regulatory mechanism influencing serum cholesterol levels and highlight the need to elucidate the genetic and enzymatic basis of this process.

The first bacterium capable of converting cholesterol to coprostanol was identified in 1973 when the *Eubacterium* strain ATCC 21,408 was isolated from rat cecum[10]. Subsequent studies led to the isolation of cholesterol-reducing bacteria from humans[11], baboons[12], and hog waste lagoons[13]. These bacteria were identified as strictly anaerobic, non-spore-forming, gram-positive bacilli and were grouped under the genus *Eubacterium* based on their shared morphological and physiological characteristics[12]. It has been hypothesized that they form a monophyletic clade because of their conserved sterol-reducing metabolism, although genetic confirmation remains impossible, as most strains were isolated before sequencing was available and have since been lost[10]. Unlike most gut microbes, cholesterol-reducing *Eubacterium* species depend on $\Delta^{5}$-3β-hydroxy steroids as essential metabolic substrates and do not proliferate in sterol-deficient environments[11]. Their metabolic activity suggests that cholesterol serves as a terminal electron acceptor, supported by their strict requirement for high cholesterol concentrations and selective biohydrogenation of the 5,6-double bond[10]. The transformation follows an indirect hydrogenation pathway, converting cholesterol to cholestenone, then to coprostanone, and finally to coprostanol[14]. This process is highly efficient, with some strains converting over 90% of the cholesterol to coprostanol[14]. Given the prevalence of *Eubacterium* species in the gut and their metabolic specialization, these bacteria are considered the primary contributors to gut microbial cholesterol biohydrogenation in mammalian hosts[15].

Identifying its cholesterol-metabolizing enzymes is essential for understanding microbial contributions to host cholesterol metabolism. *Eubacterium coprostanoligenes* ATCC 51222, the only cholesterol-reducing *Eubacterium* species with a sequenced genome and available isolates, serves as a key model for elucidating the enzymatic basis of cholesterol biohydrogenation[13]. The *ismA* gene, found in *E. coprostanoligenes* ATCC 51222, was proposed to catalyze cholesterol oxidation to cholestenone and reduction of coprostanone to coprostanol[16]. While metagenomic and biochemical evidence support its involvement, some coprostanol-positive samples lack *ismA*, suggesting that additional enzymes may contribute to this process[16].

Here, we identify and characterize SpiR (Sterol processing isomerase and Reductase), a steroid $\Delta^{5\text{-}4}$ isomerase/3-keto reductase from *Eubacterium coprostanoligenes*, which catalyzes the first and obligate step in gut microbial cholesterol metabolism. Biochemical assays reveal that SpiR oxidizes cholesterol and related steroids with high specificity and stereoselectivity, functions as an NAD(H)-dependent homodimer and exhibits stronger substrate affinity than the previously implicated enzyme IsmA. Phylogenetic analysis shows that SpiR clusters with steroid $\Delta^{5\text{-}4}$ isomerases from both prokaryotic and eukaryotic organisms. While *spiR* homologs are taxonomically restricted to a clade of uncultured *Acutalibacteraceae*, metagenomic surveys reveal that they are broadly prevalent in human gut microbiomes of cholesterol converters. Across three independent cohorts integrating metagenomic and metabolomic data, the presence of *spiR* homologs is consistently tracked with coprostanone production and fecal cholesterol depletion, including samples lacking *ismA*. Classifiers based on *spiR* outperformed *ismA*-based models in predicting cholesterol conversion (AUC = 0.95 vs. 0.81), resolving prior inconsistencies and establishing *spiR* as a more

robust biomarker. These findings redefine the enzymatic basis of microbial cholesterol metabolism and highlight SpiR as a promising mechanistic candidate and predictive marker of microbiome-mediated cholesterol conversion.

## Results

### SpiR is a putative steroid $\Delta^{5\text{-}4}$ isomerase/3-keto reductase involved in cholesterol metabolism in *Eubacterium coprostanoligenes*

To identify the key genetic components involved in cholesterol metabolism in *E. coprostanoligenes*, we investigated homologs of known steroid-transforming enzymes encoded in its genome. In our previous study, we identified a gene frequently fused with steroid 5β-reductase, which facilitated the conversion of pregnenolone to 3β,5β-tetrahydroprogesterone[17]. The conversion of pregnenolone to progesterone is catalyzed by 3β-hydroxysteroid dehydrogenase/isomerase (3β-HSDH/I). Given the close resemblance between pregnenolone-to-progesterone and cholesterol-to-cholestenone conversions, we hypothesized that if *E. coprostanoligenes* contain an enzyme homologous to 3β-HSDH/I, they may catalyze a similar reaction in cholesterol metabolism (Fig. 1a).

Through our search for homologs of 3β-HSDH/I, we identified SpiR (WP_242941742.1), which shares 39.5% protein sequence identity with the verified 3β-HSDH/I (dw0523) (Fig. 1b). SpiR is distinct from IsmA, which has been previously linked to cholesterol dehydrogenation, with only 10.7% identity between the two proteins (Fig. 1b). SpiR and IsmA belong to the SDR family, which comprises a functionally diverse group of oxidoreductases characterized by a single domain with a Rossmann fold. Other enzymes that possess the Rossmann fold and play a role in cholesterol metabolism include AcmA from *Sterolibacterium denitrificans* and Rv1106c from *Mycobacterium tuberculosis*[18,19]. These enzymes catalyze the oxidation of cholesterol to cholestenone in an oxygen-independent manner[18,19]. Although neither AcmA nor Rv1106c shares high sequence similarity (>25%) with 3β-HSDH/I or SpiR, their predicted structural similarities suggest that SpiR plays a role in cholesterol metabolism (Fig. 1c). Specifically, AcmA exhibits a high degree of structural similarity to SpiR, suggesting that SpiR performs an analogous function in cholesterol metabolism within *E. coprostanoligenes* (Fig. 1d). Given its sequence similarity to 3β-HSDH/I, high structural homology to AcmA, and presence within *E. coprostanoligenes*, we hypothesized that SpiR functions as a steroid $\Delta^{5\text{-}4}$ isomerase/3-keto reductase.

### SpiR catalyzes the conversion of cholesterol to cholestenone

We investigated whether SpiR catalyzes the conversion of cholesterol to cholestenone. To assess SpiR activity, we conducted assays using cell lysates from *E. coli* heterologously expressing SpiR incubated with cholesterol, followed by liquid chromatography-mass spectrometry (LC-MS) analysis. Prior to the activity assay, we confirmed the expression of recombinant SpiR and IsmA by SDS-PAGE. Distinct bands at the expected molecular weights of approximately 72 kDa for SpiR and 33 kDa for IsmA were observed in the induced samples but were absent in lysates from cells carrying an empty vector. These bands were also present in the post-lysis supernatant fractions, indicating that both proteins were expressed in soluble form (Fig. S1).

We utilized electrospray ionization (ESI) mass spectrometry to assay cholestenone production by transformed E. coli strains expressing SpiR and IsmA. To improve the detection of cholestenone, we derivatized it with Girard's Reagent P, as has been done in previous studies[20,21] (Fig. 2a). *E. coli* lysates containing an empty vector served as a control and showed no cholesterol conversion, whereas those expressing SpiR efficiently produced cholestenone (Fig. 2b). These results confirm that SpiR functions as an oxidoreductase acting on cholesterol to produce cholestenone. A comparison of cholestenone signals between *E. coli* lysates expressing IsmA and SpiR revealed that

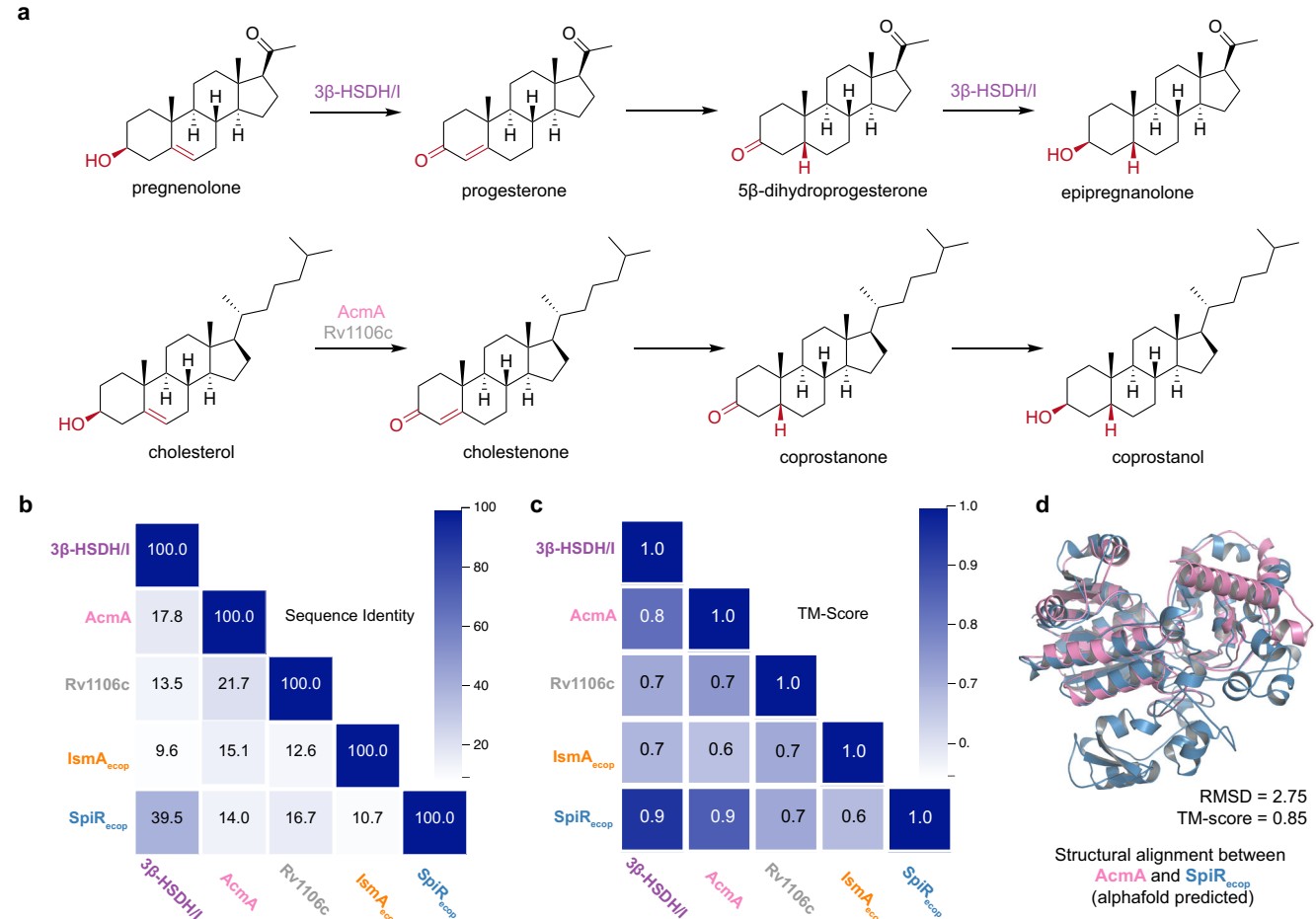

**Fig. 1 | Enzymatic pathway, sequence, and structural comparison of steroid-transforming enzymes. a** Enzymatic pathways for the conversion of pregnenolone to epipregnanolone (top), and cholesterol to coprostanol (bottom). Known enzymes catalyzing the substeps of the reaction are indicated above. **b** Pairwise sequence identity matrix of Rossmann fold-containing enzymes. Values indicate percent identity, with deeper blue coloration representing higher sequence identity. **c** Predicted structural identity matrix of enzymes that may act on cholesterol. Values indicate the TM score, with deeper blue coloration representing higher structural similarity. **d** Alphafold-predicted structural alignment of AcmA (pink) and SpiR (blue).

SpiR generated a higher cholestenone signal, with a marginally significant difference ($U = 9.0$, $p = 0.05$, Mann–Whitney U test).

## SpiR catalyzes pregnenolone-to-progesterone conversion, indicating broader steroid activity

The ability of *E. coprostanoligenes* to convert pregnenolone to progesterone has been previously reported[14]. We hypothesized that SpiR, which has been shown to catalyze cholesterol oxidation, may be responsible for this activity. To test this hypothesis, we evaluated whether SpiR catalyzes the conversion of pregnenolone to progesterone. Consistent with our observations for cholesterol, *E. coli* expressing SpiR converted pregnenolone to progesterone, whereas negative controls containing an empty vector showed no conversion (Fig. 2c). These results confirm that SpiR catalyzes 3β-hydroxysteroid oxidation and $\Delta^{5-4}$ isomerization, which are pivotal reactions in the microbial metabolism of steroid hormones and support the conclusion that SpiR plays a broader role in microbial steroid metabolism.

## SpiR reduces 3-keto steroids into stereospecific 3β-hydroxyl-steroids

To assess the enzymatic activity and stereospecificity of SpiR, we investigated its ability to catalyze the reduction of the 3-keto group in dihydroprogesterone to form tetrahydroprogesterone (Fig. 3a). Because 3-keto reduction can yield either 3α- or 3β-hydroxysteroid products, we used LC-MS with a phase column capable of resolving the four major stereoisomers: allopregnanolone (3α,5α), isopregnanolone (3β,5α), pregnanolone (3α,5β), and epipregnanolone (3β,5β).

When *E. coli* heterologously expressing SpiR was incubated with either 5α-dihydroprogesterone or 5β-dihydroprogesterone, the predominant product detected matched the retention times of the corresponding 3β-hydroxy isomers (isopregnanolone and epipregnanolone, respectively) (Fig. 3b, c). These results demonstrate that SpiR functions as a 3-keto reductase, reducing both 5α- and 5β-dihydroprogesterone to 3β-hydroxy tetrahydroprogesterone with consistent stereochemistry across divergent substrate conformations.

Together, these findings establish SpiR as a multifunctional enzyme capable of catalyzing three key transformations in microbial steroid metabolism: oxidation of the 3β-hydroxyl group in cholesterol and pregnenolone, $\Delta^{5-4}$ double-bond isomerization, and stereospecific reduction of 3-keto groups to 3β-hydroxy configurations.

## SpiR preferentially binds to cholesterol

To characterize the substrate specificity of SpiR, we evaluated its binding affinity for the key substrates involved in these transformations: cholesterol, coprostanone, pregnenolone, 5β-dihydroprogesterone, and cholestenone. Pregnenolone and 5β-dihydroprogesterone were also tested but included here primarily as comparators to cholesterol and coprostanone. Isothermal titration calorimetry (ITC) experiments revealed a higher binding affinity for cholesterol ($K_d = 539$ nM) and coprostanone ($K_d = 660$ nM)

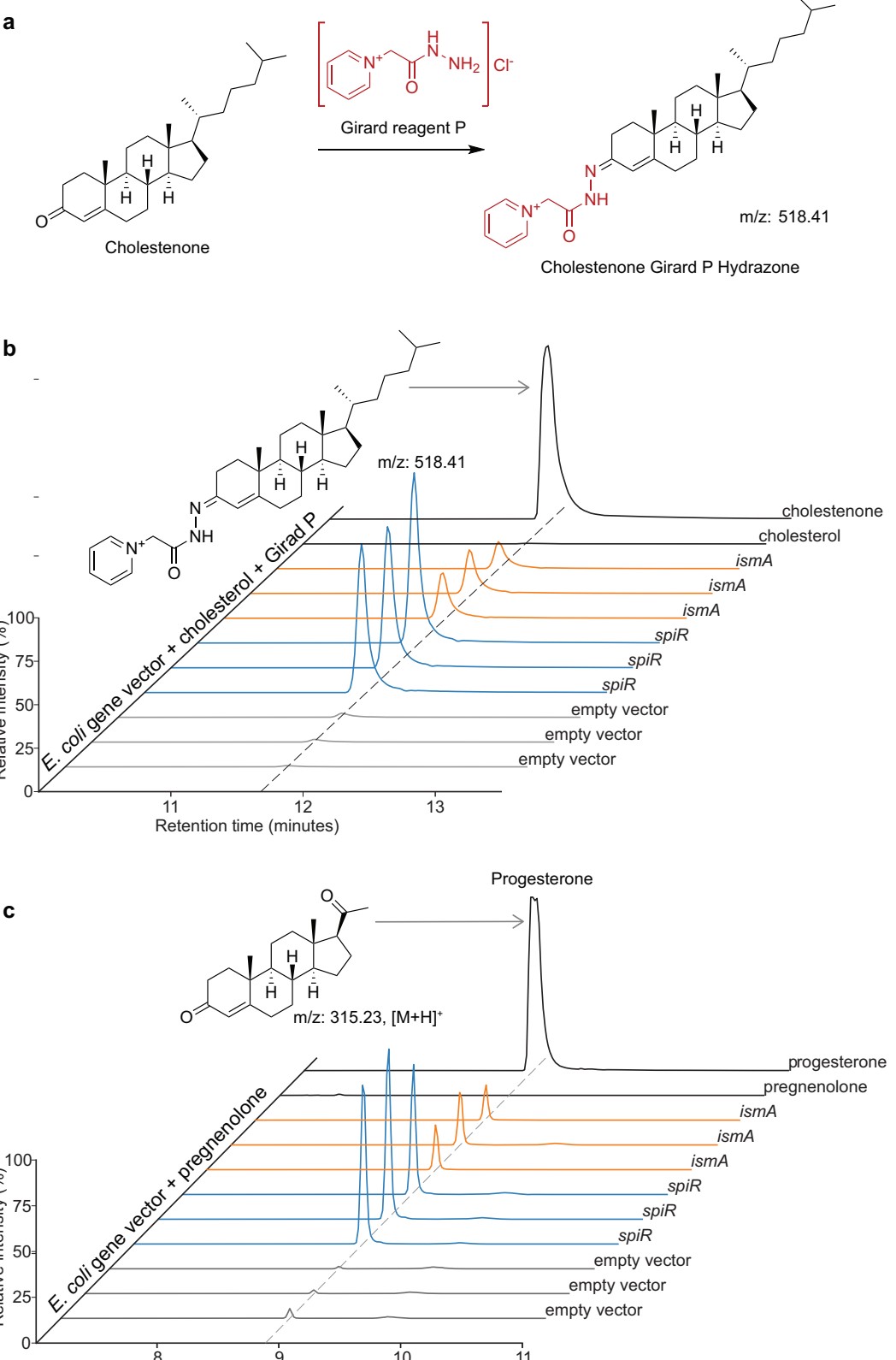

**Fig. 2 | LC-MS detection of cholesterol- and pregnenolone-transforming activity. a** Derivatization of cholestenone to cholestenone hydrazone using Girard's Reagent P. **b** Chromatogram showing the detection of cholestenone hydrazone in derivatized extracts of *E. coli* expressing a vector control (gray), *spiR* (blue), or *ismA* (orange) incubated with cholesterol. Standards of derivatized cholesterol and cholestenone are shown in black. **c** Chromatogram showing the detection of progesterone in extracts of *E. coli* expressing a vector control (gray), *spiR* (blue), or *ismA* (orange) incubated with pregnenolone. Progesterone and pregnenolone standards are shown in black. For each experimental condition, the three independent biological replicates (*n* = 3) are depicted in the graph.

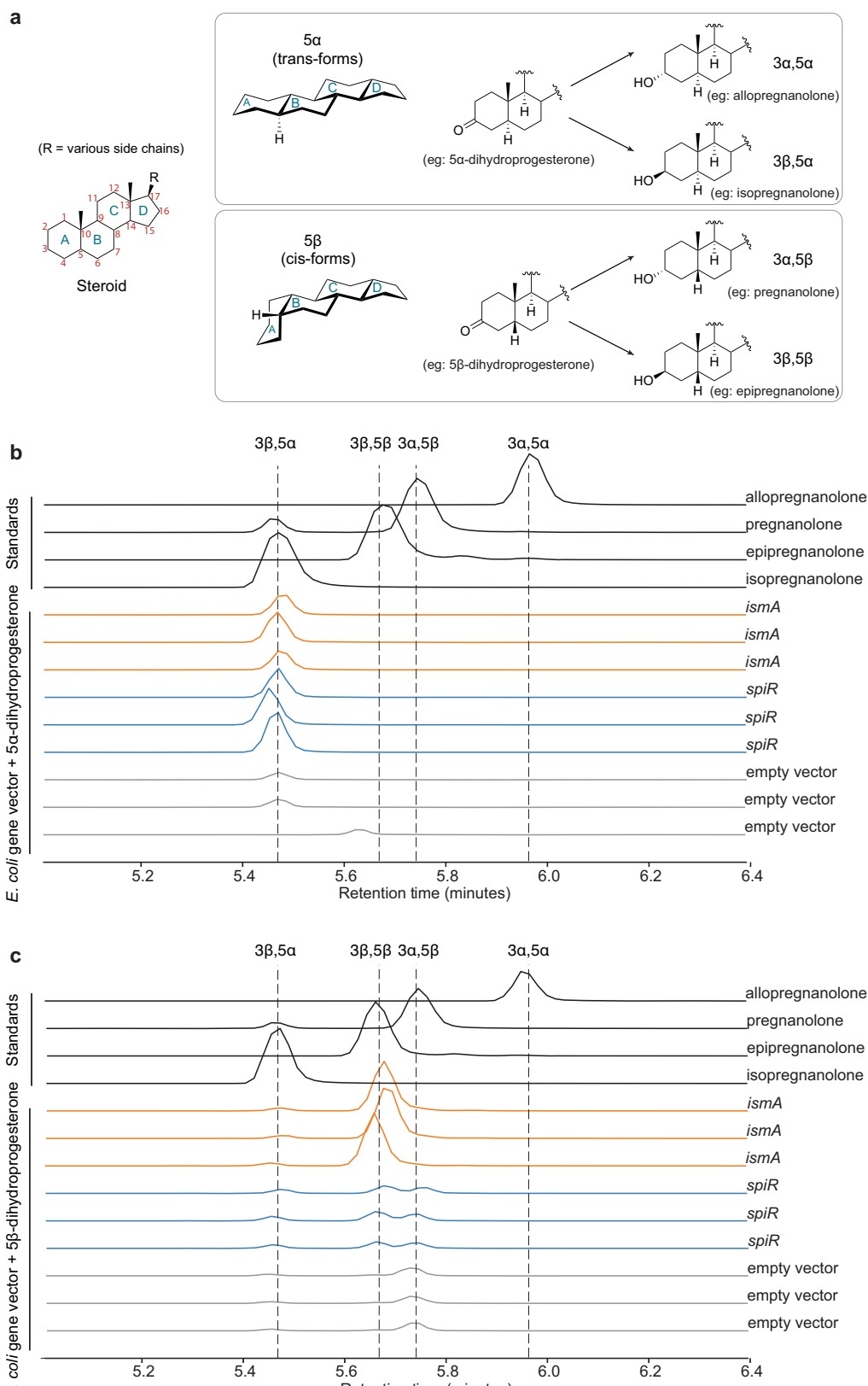

(Fig. 4b, c) than for homologous steroid hormones. Pregnenolone, which differs from cholesterol only in the absence of a hydrocarbon tail at C17, exhibited 6.12-fold weaker binding affinity (Kd = 3.3 μM) (Fig. S2). 5β-dihydroprogesterone, which lacks the same hydrocarbon tail present in coprostanone, bound SpiR with a 1.34-fold lower affinity (Kd = 887 nM) (Fig. S2). This suggests a substrate preference for coprostanone over 5β-dihydroprogesterone, although this preference is less marked than that between cholesterol and pregnenolone. These differences in binding affinity are likely the result of interactions between the enzyme and the functional groups decorating the steroid core, despite the similarities in the core structures. However, SpiR did not detectably

**Fig. 3 | LC-MS detection of stereospecific 3β-reduction of steroids. a** Structure of the steroid core showing carbon numbering and ring designations (A–D). Right: stereoisomers of tetrahydroprogesterone derived from 5α- and 5β-dihydroprogesterone, differing in hydroxyl orientation at C3 (α or β) and A/B ring fusion geometry (trans or cis). **b, c** LC-MS analysis of steroid products from *E. coli* expressing empty vector (gray), *ismA* (orange), or *spiR* (blue) incubated with (**b**) 5α-dihydroprogesterone or (**c**) 5β-dihydroprogesterone. Authentic standards (black)

define the retention times for 3α,5α (allopregnanolone), 3β,5α (isopregnanolone), 3α,5β (pregnanolone), and 3β,5β (epipregnanolone) isomers. The vertical dashed lines indicate the standard retention times. *spiR* consistently produces 3β-hydroxy isomers from both 5α- and 5β-dihydroprogesterone, identifying it as a stereoselective 3β-hydroxysteroid dehydrogenase. For each experimental condition, the three independent biological replicates (*n* = 3) are depicted in the graph.

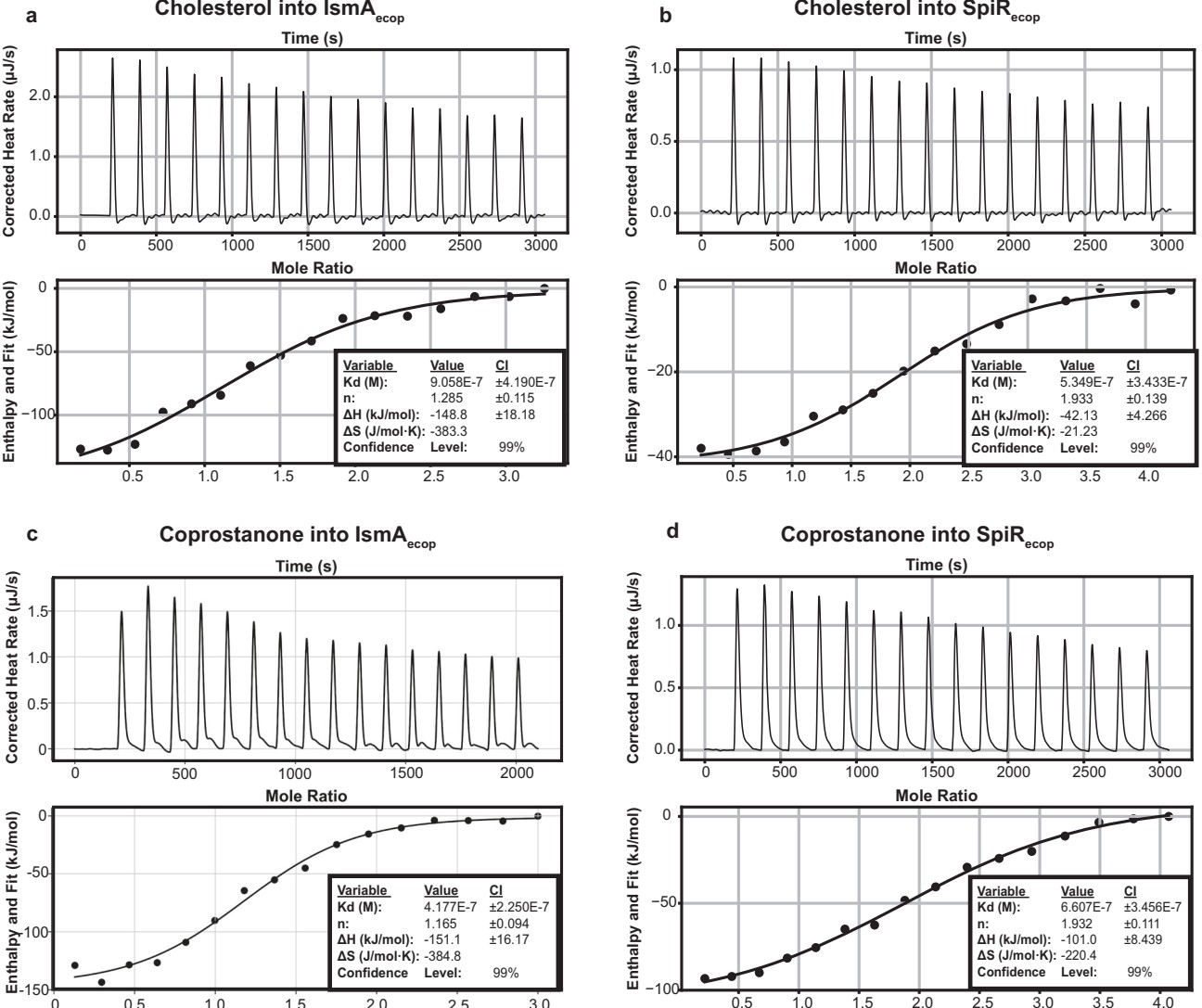

**Fig. 4 | Steroid binding to IsmA and SpiR.** Binding of IsmA (**a**) and SpiR (**b**) to cholesterol and binding of IsmA (**c**) and SpiR (**d**) to coprostanone as measured by ITC. Raw data (heats of binding) are shown in the upper graph of each panel, whereas integrated binding isotherms with their independent binding model fitting are shown in the lower panel. Recombinant IsmA and SpiR were placed in the calorimeter cell, and the experiment was performed at 25 °C with successive 3.1 μL injections of the ligand. The binding of all the traces was performed using a 50 μm ligand and 5 μm protein. The thermodynamic parameters with standard errors were calculated based on each curve using the NanoAnalyze Software package.

bind to cholestenone, consistent with its role in product release following cholesterol oxidation (Fig. S2). These results indicate that cholesterol and coprostanone are the preferred ligands of SpiR, underscoring their likely role in gut microbial cholesterol metabolism.

As IsmA was also found to metabolize cholesterol, we compared its cholesterol-binding affinity to SpiR. IsmA titrated with cholesterol showed 1.68-fold lower binding affinity than that observed between SpiR and cholesterol (Kd = 908 nM) (Fig. 4a). This comparatively lower affinity suggests that even though IsmA may participate in cholesterol

metabolism, SpiR is likely the more specialized enzyme for cholesterol reduction in the gut. IsmA also bound coprostanone (Kd = 417 nM) (Fig. 4c) but did not bind to cholestenone (Fig. S2).

To further characterize the likely enzyme cofactors, we tested their binding to NADH, NADPH, and NAD$^+$. SpiR was able to bind to both NADH (Kd = 1.03 μM) with *n* = 1.813 ± 0.140 and NAD$^+$ (Kd = 1.14 μM) with *n* = 1.974 ± 0.120, exhibiting a comparable binding affinity for NADH and NAD$^+$ as cofactors (Fig. S3). However, no binding was observed between SpiR and NADPH (Fig. S3). Both substrate and cofactor binding were further supported by NADH/NAD$^+$ oxidation-

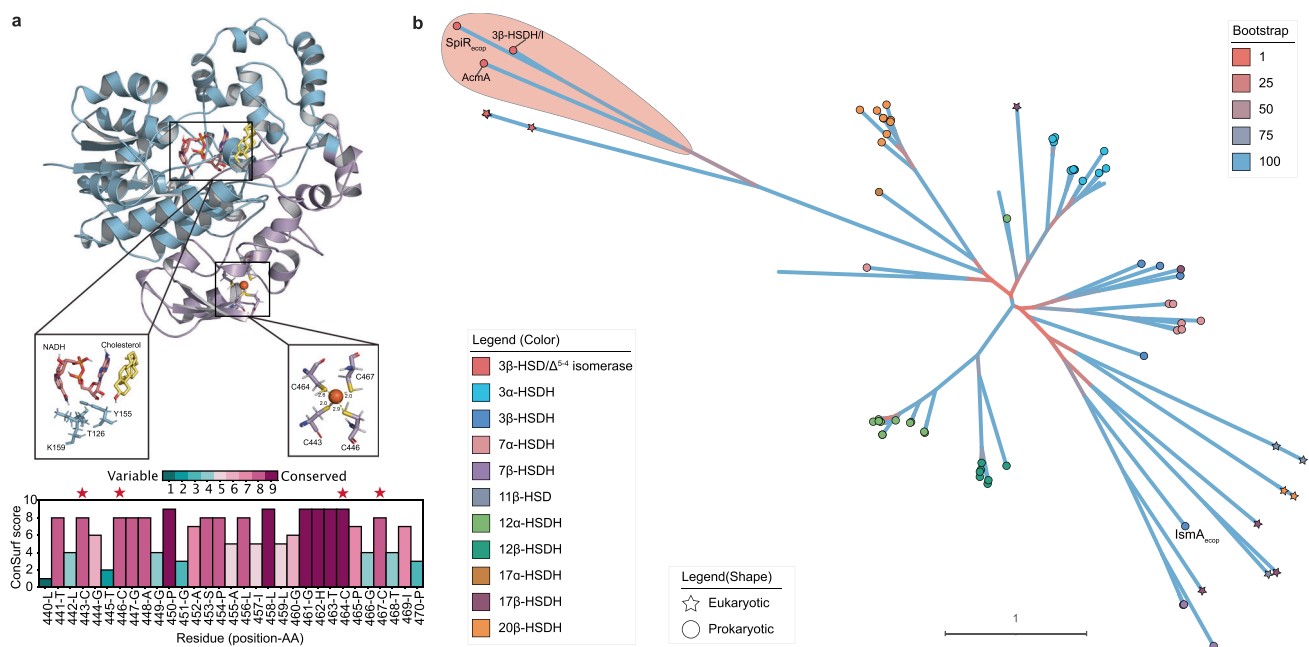

**Fig. 5 | Structural conservation and phylogenetic reconstruction of SpiR. a** The predicted AlphaFold2 structure of SpiR. The C-terminal domain is colored in purple. Putative active-site residue interactions with docked cholesterol (yellow) and NADH (pink) are shown, as predicted by Autodock vina. The putative rubredoxin Fe(Cys)₄ site is shown with distances to the Fe(II) ion (in Å), and ConSurf scores for the cysteines at this site are plotted. **b** Phylogenetic tree representing evolutionary relationships between HSDH enzymes. Branch colors indicate bootstrap support, with cooler colors (e.g., blue) representing higher support and warmer colors (e.g., red) indicating lower support. Tip shapes differentiate taxonomy, where circles represent prokaryotic sequences and stars denote eukaryotic sequences. Tip labels are color-coded based on the hydroxyl functional group position on steroid substrates, where HSDH enzymes act, as well as their stereochemistry, distinguishing enzymes acting on different steroid hydroxyl positions and configurations.

reduction assays, which confirmed that SpiR and IsmA are biologically active and reinforced ITC results demonstrating that SpiR binds NADH and NAD⁺ (Fig. S4). These results align with the cofactors bound to similar reductase enzymes and validate our prediction of the cofactors necessary for the oxidation and reduction of enzyme substrates[22–24].

Cholesterol, coprostanone, pregnenolone, and 5β-dihydroprogesterone exhibited stoichiometric values of $1.933 \pm 0.139$, $1.932 \pm 0.111$, $1.970 \pm 0.263$, and $1.909 \pm 0.149$, for binding, respectively (Figs. 4b, c and S2). These values indicate that either two ligand molecules bind to a single enzyme monomer or that the enzyme primarily exists as a dimer. Some enzymes can accommodate two substrates within the same binding site owing to π–π interactions within the aromatic core[25]. To investigate the oligomeric state of SpiR, a dynamic light scatterer (DLS) analysis was performed. The measured hydrodynamic radius was 3.76 nm, which is consistent with the theoretical radius of its dimeric state (3.68 nm) and larger than the expected radius of its monomeric form (2.94 nm)[26]. This indicates that SpiR is predominantly present as a dimer in the solution (Fig. S5). Given that other hydroxysteroid dehydrogenase (HSDH) families have also been observed to form homodimers[27,28], the stoichiometry and DLS results suggest that SpiR functions as a homodimer.

## SpiR clusters phylogenetically with known steroid Δ⁵⁻⁴ isomerases within the SDR superfamily

The structural similarities between steroid substrates and their stoichiometric values suggest a conserved model of ligand-SpiR binding; therefore, we sought to identify its catalytic residues through structural modeling and ligand-docking simulations. Using the AlphaFold2-predicted structure of SpiR, we docked NAD⁺ with cholesterol as a representative steroid substrate to define the putative active site. This analysis revealed that three residues, T126, Y155, and K159, are in close proximity to the docked ligands (Fig. 5a). These

residues correspond to the canonical catalytic triad found in the SDR superfamily, in which tyrosine typically acts as a general acid/base catalyst, lysine modulates pKa through hydrogen bonding as a general base, and threonine as a proton donor[29].ConSurf analysis and multiple sequence alignment confirmed that these residues are highly conserved across SpiR homologs (Fig. 5a). These findings suggest that SpiR uses a conserved SDR mechanism, with T126, Y155, and K159 forming the catalytic triad that is required for cholesterol oxidation.

Unlike canonical SDR enzymes, which are typically 250–250 amino acids in length, SpiR is extended in length with 520 amino acids and contains an additional C-terminal domain. Within this domain, four cysteine residues (C443, C446, C464, and C467) are conserved across homologs, suggesting strong evolutionary pressure to retain this feature. Structural analysis demonstrated that the cysteines adopt a compact tetrahedral configuration, with sulfur–sulfur separations consistently in the range of ~3.5–3.9 Å. This arrangement is incompatible with [2Fe–2S] or [4Fe–4S] clusters, which require irregular cysteine spacing and bridging sulfides. Although a Zn(Cys)₄ site could, in principle, account for the observed geometry, such motifs typically serve structural rather than catalytic functions, are not universally conserved, and are therefore unlikely in this context. Taken together, the conservation and stereochemistry of these residues are most consistent with coordination of a rubredoxin-like Fe(Cys)₄ center. To our knowledge, this represents a previously unrecognized feature within the otherwise metal-independent SDR superfamily[30].

To place SpiR in an evolutionary context, we compiled a curated dataset of biochemically characterized HSDHs. This included representative enzymes from previously published surveys of microbial HSDH diversity[28], as well as additional analyzed sequences of interest, including SpiR, AcmA, and related homologs. To broaden the functional and taxonomic scope of the phylogenetic analysis, we also

incorporated additional enzymes not featured in the prior study, such as 17β-HSDH (DesG), 17α-HSDH (DesF), and 3β-HSDH/I[17,31,32]. By comparing SpiR with other HSDH enzymes, we elucidated the evolutionary context of the enzyme. Phylogenetic analysis revealed extensive diversity among the HSDH enzymes, indicating multiple evolutionary trajectories for steroid metabolism. SpiR, 3β-HSDH/I, and AcmA form a distinct clade that is a sister group to the eukaryotic 3β-HSD/$\Delta^{5-4}$ isomerase, suggesting a close evolutionary relationship between these prokaryotic and eukaryotic enzymes. AcmA catalyzes the conversion of cholesterol to cholestenone through $\Delta^{5-4}$ isomerization, a function that is also carried out by eukaryotic 3β-HSD/$\Delta^{5-4}$ isomerase. This clade is phylogenetically distant from the rest of the HSDH enzymes, including IsmA, which belongs to a more divergent and functionally distinct group. The separation of SpiR and its related enzymes from other HSDHs involved in bacterial sterol metabolism suggests distinct evolutionary pressures shaping their functions, with SpiR aligning closely with the known cholesterol isomerase, AcmA.

## SpiR homologs are encoded by uncultured Acutalibacteraceae

We performed a survey of the *spiR*-like gene family and identified them exclusively in bacteria from the Clostridia class and Oscillospirales order in the Acutalibacteraceae family (Fig. 6a). Acutalibacteraceae are prevalent in anoxic gut environments and are linked to bile acid metabolism in the large intestine, where they play a role in deconjugation and transformation pathways[20]. The Acutalibacteraceae family is primarily represented by uncultured genera, consisting of nearly all genomes derived from metagenomes (96%, 1096 out of 1144), suggesting that much of the diversity within this family and the cholesterol-reducing microbial community has not been explored. Among the identified species, 308 contained *spiR* homologs and 180 possessed *ismA* homologs. Most *spiR* homologs (314/317) were encoded by species in one Acutalibacteraceae clade, and 148 species within this clade encoded both *ismA* and *spiR* homologs, indicating a significant overlap between the distribution of both gene types (Fig. 5b). The restricted taxonomic distribution and frequent co-occurrence of *spiR* homologs and *ismA* homologs suggest that this clade represents a niche-adapted lineage specialized for cholesterol metabolism in the anaerobic gut, consistent with earlier reports describing non-spore-forming, strictly anaerobic *Eubacterium* species that depend on sterols as essential electron acceptors[10,12].

To further validate this taxonomic pattern, we characterized two cholesterol-reducing isolates capable of converting cholesterol to coprostanol. The first was a human-derived bacterium, *Astrobacillus crystallinus* SVS042 (accession JBPPLN000000000), isolated from the feces of a healthy adult. This strain reduced cholesterol to coprostanol and was placed by phylogenomic analysis within the *Acutalibacteraceae* genus *CAG-177*[33]. Genome sequencing revealed the presence of *spiR* but the absence of *ismA*, suggesting that *ismA* is not strictly required for cholesterol conversion in the human gut. The discovery of a human-derived, *spiR*-positive cholesterol reducer expands the known host range of this metabolic capacity beyond the previously characterized swine-and rat-associated isolates[10,34]. We also sequenced the genome of *Eubacterium* ATCC 21,408 (accession SAMN50847228), a rat-associated cholesterol-reducing bacterium originally described by Eyssen and colleagues[12,34]. Phylogenomic analysis placed this strain within the genus *Fimenecus* of the *Acutalibacteraceae* family. Similar to related taxa, *Eubacterium* ATCC 21,408 encodes both *spiR* and *ismA* homologs (Supplementary Data 2), providing an independent example of their co-occurrence within a single lineage. Together, these two sequenced isolates, one human and one rat-associated, corroborate that cholesterol reduction to coprostanol is a conserved metabolic trait within the *Acutalibacteraceae* and reveal the presence of *spiR*-encoded reductases mediating cholesterol conversion across diverse mammalian hosts.

## *spiR* homologs strongly correlate with cholesterol conversion across human gut microbiome cohorts

Many species encode both *spiR* homologs and *ismA* homologs, raising the possibility that previous associations between *ismA* and cholesterol conversion may have been confounded by *spiR*. Kenny et al.[16] found coprostanol in samples lacking *ismA*, implying that other enzymes may contribute. We hypothesized that *spiR* homologs may underlie this association and evaluated their link to cholesterol conversion. To evaluate this hypothesis, we analyzed paired metagenomic and metabolomic datasets, including the two cohorts used by Kenny et al.[16]: HMP2 ($n = 400$)[35] and PRISM ($n = 151$)[36], and included an additional cohort, PROTECT ($n = 90$)[36,37].

Our results suggest that *spiR* homologs are more strongly enriched in samples with detected cholesterol conversion than *ismA* homologs. Samples were grouped based on the presence (≥1 CPM) or absence of *spiR* and *ismA* homologs in fecal metagenomes, as well as the detection of cholesterol conversion in the fecal metabolome. Consistent with previous studies[1,9,38,39], we found that the samples had high variability in cholesterol conversion, with the overall count of converters (325 individuals) roughly equal to that of non-converters (308 individuals) (Fig. 6b). There was a significantly higher prevalence of *spiR* homologs in converter samples than in non-converters across all three cohorts, as determined by one-sided Fisher's exact tests. In HMP2, the odds ratio was 43.57 (95% CI: 25.83−∞, $P = 1.80 \times 10^{-53}$); in PRISM, 408.18 (95% CI: 97.39−∞, $P = 1.36 \times 10^{-34}$); and in PROTECT, 55.11 (95% CI: 13.98−∞, $P = 1.54 \times 10^{-12}$). On average, 89.7% of converters harbored *spiR* homologs compared to only 6.18% of non-converters. Although *ismA* homologs were also significantly enriched in converter samples compared to non-converters, as determined by one-sided Fisher's exact tests (HMP2: OR = 22.16, 95% CI: 12.30−∞, $P = 6.29 \times 10^{-31}$; PRISM: OR = 109.80, 95% CI: 20.79−∞, $P = 7.74 \times 10^{-18}$; PROTECT: OR = 26.39, 95% CI: 6.85−∞, $P = 2.96 \times 10^{-8}$), the overall prevalence was lower than that of *spiR* homologs, with *ismA* homologs detected in only 52.2% of cholesterol converters on average. In addition, these trends were not confounded by disease status, with neither *spiR* homologs nor *ismA* homolog presence differing significantly across ulcerative colitis (UC), Crohn's disease (CD), and non-inflammatory bowel disease (non-IBD) samples (Fig. S6). These patterns suggest that the presence of genes, not diseases, is the primary contributor to the observed differences in cholesterol metabolism.

Moreover, the presence of the *spiR* homologs was more strongly associated with increased levels of cholestenone and coprostanone in stool than *ismA* homologs across all three cohorts (HMP2, PRISM, and PROTECT). *spiR* homolog presence yielded robust strong associations with coprostanone (HMP2: $P = $ U = 2614, $2.6 \times 10^{-49}$; PRISM: U = 133, $P = 1.3 \times 10^{-24}$; PROTECT: U = 79, $P = 2.8 \times 10^{-13}$), exceeding those for *ismA* homologs (HMP2: U = 4270, $P = 6.1 \times 10^{-32}$; PRISM: U = 249, $P = 1.7 \times 10^{-20}$; PROTECT: U = 127, $P = 3.4 \times 10^{-10}$) (Fig. 6c). Cholestenone followed the same trend, where those with *spiR* homologs tended to have increased cholestenone levels (one-sided Mann–Whitney U test, U = 8214, 300, 163 and $P = 1.9 \times 10^{-22}$, $5.9 \times 10^{-22}$, and $3.9 \times 10^{-11}$, respectively) than those without the gene, whereas the corresponding P-values for *ismA* homologs were consistently higher (weaker associations): U = 9566, 743, 188 and $P = 2.5 \times 10^{-11}$, $1.3 \times 10^{-13}$, and $9.5 \times 10^{-9}$, respectively (Fig. 6d). In addition, *spiR* homologs and *ismA* homologs were both associated with significantly reduced stool cholesterol concentrations in HMP2 and PRISM, but not in PROTECT (*spiR* homologs: HMP2: U = 26372, $P = 1.4 \times 10^{-11}$; PRISM: U = 3702, $P = 0.002$; PROTECT: U = 960, $P = 0.44$; *ismA* homologs: HMP2: U = 24614, $P = 5.8 \times 10^{-16}$; PRISM: U = 4276, $P = 2.3 \times 10^{-10}$; PROTECT: U = 911, $P = 0.19$) (Fig. 6e).

We extended this analysis to the species level by examining the presence of species encoding *spiR* homologs (*spiR*+ species) and species encoding ismA homologs (*ismA*+ species) in metagenomic samples. We applied an approach similar to that of Kenny et al. to assess

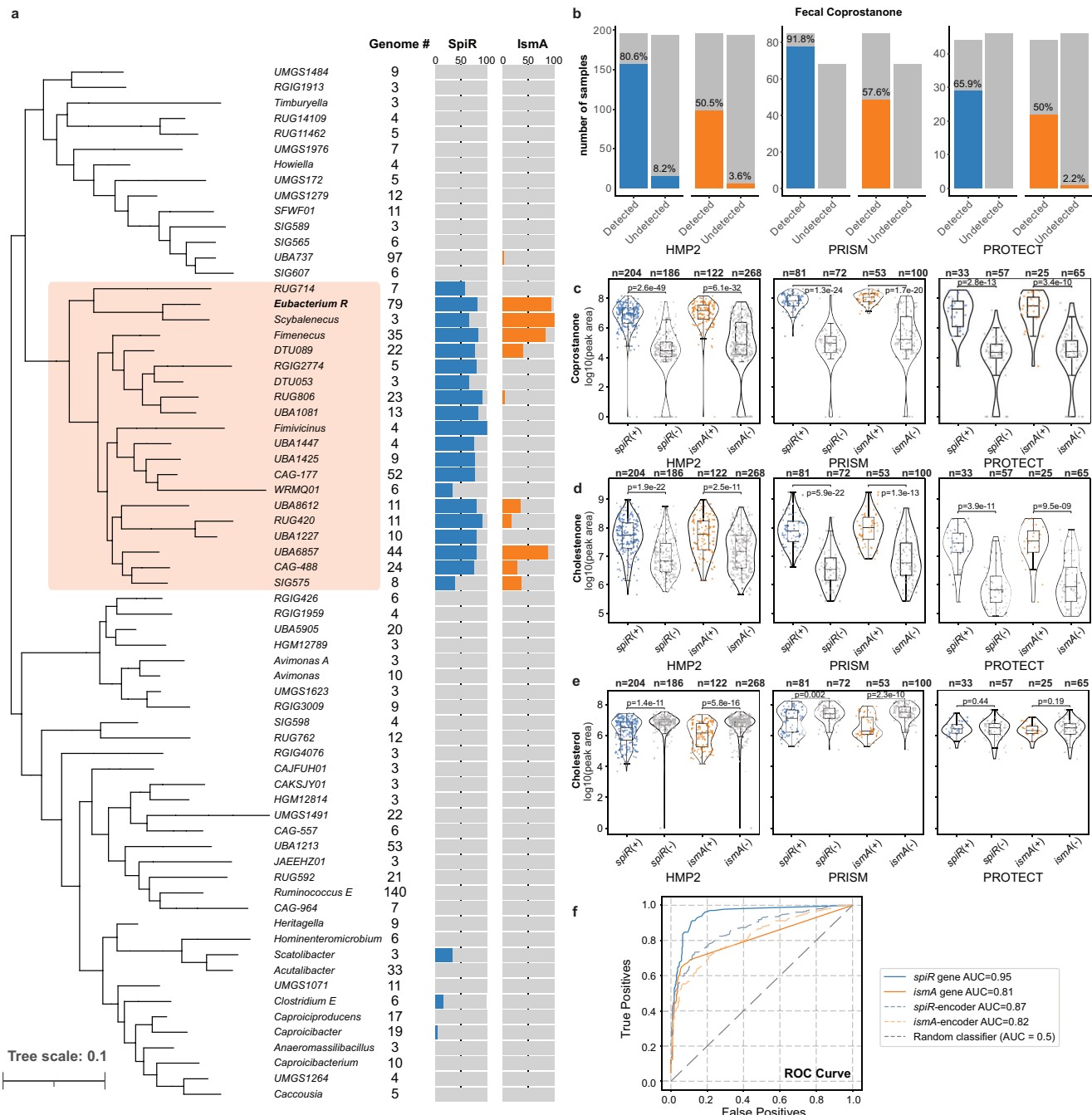

**Fig. 6 | Genus distribution and presence of *spiR* in human gut metagenomes.**
**a** Genus-level phylogeny of the family Acutalibacteraceae showing the number of genomes and percentage of species with *spiR* homologs and *ismA* homologs per genus. A clade containing species with both *spiR* homologs and *ismA* homologs features is highlighted in orange. The genus-level tree was generated by pruning the GTDB species tree using the gotree prune command and then collapsing the species by genus. Genera with fewer than three species were removed.
**b** Investigation of the association between *spiR* homologs, *ismA* homologs, and coprostanone formation in three paired human metagenomic and metabolomic datasets (HMP2, PRISM, and PROTECT). The presence of *spiR* was more strongly associated with coprostanone detection than *ismA*. **c** The presence of *spiR* homologs or *ismA* homologs is associated with lower stool cholesterol levels. Each point represents a paired metabolomics–metagenomic sample. The *y*-axis represents the log10 area of the identified cholesterol peak area as a proxy for concentration. *P*-values represent the results of a one-sided Mann–Whitney U test

(alternative = greater) comparing the cholesterol levels of samples with and without *spiR*. *n* represents the number of metagenomic samples analyzed for each group within each cohort. The lines in the boxplot represent the median and IQR. (**d**), **e** The presence of *spiR* homologs was more strongly associated with lower stool **d** cholestenone and **e** coprostanone levels than *ismA*. *P*-values represent the results of a one-sided Mann–Whitney U test (alternative = less for coprostanone and cholestenone, and alternative = greater for cholesterol) comparing the cholesterol levels of samples with and without *spiR*. **f** Receiver operating characteristic curve (ROC) for the prediction of coprostanone detection in stool samples from all three cohorts (HMP2, PRISM, and PROTECT) based on the presence of *spiR* homologs, *ismA* homologs, *spiR*-encoding species, or *ismA*-encoding species with varying CPM thresholds. The presence of *spiR* homologs (AUC = 0.95) showed the best predictive performance compared with the other curves. The ROC curve and AUC values were calculated using the Python package scikit-learn (v1.6.1).

species presence in relation to cholesterol conversion. We found similar trends compared to the gene-level analysis, where the *spiR*+ species were more strongly associated with cholesterol conversion than the *ismA*+ species (Fig. S7a), and higher levels of coprostanone (Fig. S7b), cholestenone (Fig. S7c), and lower cholesterol levels (Fig. S7d) in all three studies. We then classified the species into four groups based on the presence of *ismA* and *spiR* homologs. Importantly, among the converters, there were very few species with only *ismA*, whereas most species had either *spiR* only or both *spiR* and *ismA* homologs (Fig. S8). This suggests that *spiR*+ species account for most of the cholesterol conversion activity previously attributed to *ismA* and cholesterol converters, which lack *ismA* homologs.

We evaluated the predictive power of *ismA* and *spiR* homolog presence in relation to fecal coprostanone production and performed receiver operating characteristic (ROC) curve analysis at both the gene and species (encoder) levels (Fig. 6f). The presence of the *spiR* homolog alone yielded the highest classification performance, with an area under the curve (AUC) of 0.95, outperforming the *ismA* homologs (AUC = 0.81). Similarly, classifiers based on *spiR*+ species achieved higher performance (AUC = 0.87) than those based on *ismA*+ species (AUC = 0.82). These results confirm that *spiR* homologs are a stronger and more consistent predictor of coprostanone production than *ismA* homologs, whether assessed directly at the gene level or inferred from the presence of the encoding species. To test whether the co-occurrence of *ismA* and *spiR* enhances predictive power, we performed a similar ROC curve analysis on combinations of *spiR*+ and *ismA*+ species (Fig. S8). These analyses reveal that *spiR*+ alone offers the best predictive performance, and that accounting for *ismA* co-occurrence does not improve predictive power. This reinforces the conclusion that *spiR* homologs play a central role in cholesterol metabolism in the gut microbiome, and that *ismA* likely does not synergize with *spiR* in mediating cholesterol conversion.

## Discussion

This study identified SpiR as a gut bacterial steroid $\Delta^{5\text{-}4}$ isomerase/3-keto reductase that catalyzes the obligate first step and the last step in the microbial conversion of cholesterol to coprostanol. Previous work implicated IsmA in this transformation. However, across the three independent cohorts, *spiR* homologs were more predictive of the phenotype than *ismA* homologs, showing stronger associations with cholesterol conversion, coprostanol levels, and fecal sterol profiles.

These findings challenge the current *ismA*-centric model of gut microbial cholesterol metabolism and establish SpiR as a key enzymatic contributor to this pathway. Although the determination of catalytic efficiency (kcat/Km) for SpiR and IsmA would provide a more quantitative comparison of their enzymatic contributions, reliable steady-state turnover measurements of these enzymes were not feasible in this study due to technical limitations. We acknowledge this as a limitation of the present study. Even so, the substrate affinity measurements, multi-cohort enrichment analyses, and LC-MS data consistently support SpiR as the key gut microbial enzyme catalyzing cholesterol conversion.

Compared to many other nutrients in the gut, cholesterol metabolism presents challenges owing to its hydrophobicity and inability to passively diffuse across cellular membranes[40–42]. Cholesterol is amphipathic, with a tetradecahydro-cyclopentaphenanthrene ring system and poor solubility. As the activating enzyme in the cholesterol reduction pathway, SpiR catalyzes the obligate initiating oxidation of cholesterol to cholestenone, forming the electron-withdrawing group required for the next step, the ene-reduction carried out by a 5β-reductase. Together, this initial oxidation and the subsequent reductions enable the overall disposal of two electrons to cholesterol. In the anaerobic gut, where electron acceptors are scarce, the utilization of cholesterol as a terminal electron acceptor could be highly advantageous[34,43,44]. Our findings revealed that this capacity is likely

phylogenetically restricted to a clade within the Acutalibacteraceae family, suggesting that it represents a rare trait driving niche adaptation in the gut. This clade therefore represents an ideal target for identifying other lineage-specific genes involved in cholesterol metabolism, including the yet-unidentified 5β-reductase and potential cholesterol transport systems that enable substrate uptake.

In addition to this reductive pathway, recent studies have shown that *Bacteroides thetaiotaomicron* can sulfonate cholesterol via a dedicated sulfotransferase system, producing cholesterol sulfate that influences host immune and metabolic processes[45,46]. This shows that gut microbes employ multiple strategies to transform cholesterol, each with its own potential impact on host physiology and microbial ecology. Moreover, inter-microbial interactions—such as competition for cholesterol or cross-feeding between organisms with complementary transformations—are likely to further shape the metabolic fate of cholesterol in the gut. Understanding how *spiR*-encoding organisms interact with other cholesterol-transforming microbes will be essential for mapping the broader ecological and physiological impact of these pathways.

Identification of *spiR* provides a critical entry point for linking microbial cholesterol metabolism to host physiology. The genes characterized in this study provide a likely route for the microbiome to influence host cholesterol levels, but further work is needed before any robust conclusions can be made. Systemic cholesterol levels are strongly influenced by diet, bile acid metabolism, hepatic synthesis, and host genetics, and further work is required to disentangle the specific contribution of microbial pathways. Translational applications such as probiotics must therefore await further delineation of the complete pathway and further characterization of the impact of SpiR on cholesterol conversion. Future studies using gnotobiotic animal models, dietary interventions, and cultivation or synthetic biology approaches to reconstruct the complete coprostanol pathway will be critical for determining whether *spiR* exerts a causal effect on serum cholesterol levels.

If such causal links are established, *spiR*-encoding species could represent promising candidates for next-generation probiotics designed to enhance cholesterol excretion[16,47]. Future work should aim to isolate and study these organisms. Moving toward this goal will require isolating and characterizing *spiR*-positive taxa, as well as testing whether dietary interventions can selectively enrich them as a potential prebiotic strategy. For example, plant-derived sterols are metabolized by *Eubacterium* species and have been shown to influence both microbiota composition and host lipid profiles[48]. The precise delineation of the microbial genes and lineages responsible for cholesterol-to-coprostanol conversion in the gut lays the groundwork for exploring translational applications, while maintaining appropriate caution until causal roles are firmly established.

## Methods

### Cloning and transformation

Plasmids containing the genes of interest (*ismA* and *spiR*) were acquired from GenScript (Genscript.com). Plasmids were individually transformed into T7 Express *lysY*/*I*$^q$ Competent *E. coli* (New England Biolabs, C3013I) using the manufacturer's protocol. The cells were plated on Luria-Bertani (LB, Sigma-Aldrich, L3022) agar plates with 50 µg/mL kanamycin (BioBasic, 70560) to select for successfully transformed colonies. The transformed colonies were sequenced and verified using Plasmidsaurus (Plasmidsaurus.com).

### Whole cell assays

Transformed *E. coli* strains were inoculated from a glycerol stock into 250 mL of LB media with 50 µg/mL kanamycin and shaken at 32 °C overnight. The bacteria were pelleted by centrifugation at 3260 × g for 7 min and transferred to the anoxic chamber. The cell pellet was resuspended in 100 mL of anoxic LB medium with 10 µg/mL steroid,

50 μg/mL kanamycin, and 400 μM isopropyl-ß-D-1-thiogalactopyranoside (IPTG, Goldbio, 22110512481) and incubated anoxically at 37 °C for 48 h.

After incubation in the anoxic chamber, the samples were pelleted by centrifugation at $3260 \times g$ for 5 min. Three milliliters of chloroform (Fisher Chemical, C298) were added to the supernatant, and the products were separated via organic extraction in a separatory funnel. The resulting chloroform-steroid solution was air-dried until the chloroform was fully evaporated, and the extract was resuspended in 1 mL of methanol (Fisher Chemical, 232699).

### Lysate assay

Ten milliliters of overnight *E. coli* culture were seeded in 500 mL of LB medium with 50 μg/mL kanamycin and shaken at 32 °C until an $OD_{600}$ of 0.5–0.6 and 400 μM IPTG was added to the media. The bacteria were shaken overnight at 25 °C, pelleted by centrifugation at $3260 \times g$ for 10 min, and frozen at −20 °C. Pellets were thawed on ice for 30 min before the addition of pH-adjusted cell lysis buffer (20 mM Tris, 0.2 M sodium chloride), lysozyme (Research Products International, L38100), 100 μL beta-mercaptoethanol (BME, G Biosciences BC98), and protease inhibitor (Sigma-Aldrich, 04693124001). Samples were sonicated at 4 °C using a Branson 550 sonicator set to 40% amplitude, with a cycle of 15 s on and 30 s off, for a total sonication time of 2 min. The lysate was centrifuged at $14,000 \times g$ for 10 min, and the supernatant was incubated with 100 μL of 10 mM cholesterol in ethanol (IBI Scientific, IB15721), 10 μL of 100 mM $NAD^+$ (Sigma-Aldrich, 10127981001) in water, and 10 μL of 100 mM $NADP^+$ (Sigma-Aldrich, 10128031001) in water at 4 °C for 48 h. Prior to addition to the lysate, cholesterol was solubilized using methyl-β-cyclodextrin (Sigma-Aldrich, C4555). The products were extracted using chloroform and resuspended in methanol for LC-MS analysis.

### Derivatization and LC-MS

After chloroform extraction, the samples were derivatized with Girard's Reagent P (Cayman Chemical Company, 601541). 40 μL of the sample was diluted 10-fold in distilled water, and three volumes of 2 mg/mL Girard's Reagent P in methanol containing 1% acetic acid (LabChem, LC101003) were added. The samples were incubated overnight in the dark at 37 °C.

Samples were analyzed with the Bruker Maxis-II QTOF, an ultra-high-resolution Q-TOF mass spectrometer coupled with a Waters Acquity I-Class PLUS LC system. Liquid chromatography separation was performed on a Phenomenex Kinetex C18 100 Å LC Column (100 × 3 mm, 2.6 μm particle size) with mobile phase A (water with 0.1% formic acid) and mobile phase B (acetonitrile with 0.1% formic acid) according to the Oxysterol Derivatization MaxSpec® Kit (Cayman Chemical, 601540) manufacturer's protocol. The gradient program began at 2% B for 30 s, increased to 15% B at 1 min, and was maintained for 9 min before increasing to 80% B. From 10.2 min to 13 min, B was increased to 95% and then decreased to 2% for an additional 3 min. The column was maintained at 25 °C. The injection volume was 5 μL. Mass spectra were acquired under positive electrospray ionization with an ion spray voltage of 4500 V. The source temperature (dry gas) was set at 220 °C at a flow rate of 5.0 L/min. Mass spectrometric analysis was performed on a Bruker maXis II Q-TOF instrument controlled by Bruker otofControl (v5.2) and Bruker HyStar (v5.1.8.1). Extracted ion chromatograms were generated at theoretical $[M + H]^+$ m/z values using ±0.05 Da tolerance for chromatographic profiling and 10 ppm for targeted detection. Data processing was performed in Python (v3.10.12) using pyOpenMS (v3.1.0) and pyteomics (v4.7.5).

### Phase column LC-MS

To separate isomers (5α-and 5β-dihydroprogesterone), samples were analyzed with the Bruker Maxis-II QTOF, an ultra-high-resolution Q-TOF mass spectrometer coupled with a Waters Acquity I-Class PLUS LC system. Liquid chromatography separation was performed on a Phenomenex Kinetex F5 100 Å LC Column (100 × 3 mm, 2.6 μm particle size) with mobile phase A (water with 0.1% formic acid) and mobile phase B (acetonitrile with 0.1% formic acid). The gradient program began with 10% B for 2 min, increased to 55% B over the next minute, and further increased to 65% B for 11 min. B was increased to 90% over the next minute, maintained at 90% B for 2 min, decreased to 10% over the course of 1 min, and maintained for 3 min. The column was maintained at 45 °C. The injection volume was 5 μL. Mass spectra were acquired under positive electrospray ionization with an ion spray voltage of 4500 V. The source temperature (dry gas) was set to 220 °C with a flow rate of 5.0 L/min. Mass spectrometric analysis was performed on a Bruker maXis II Q-TOF instrument controlled by Bruker otofControl (v5.2) and Bruker HyStar (v5.1.8.1). Extracted ion chromatograms were generated at theoretical $[M + H]^+$ m/z values using ±0.05 Da tolerance for chromatographic profiling and 10 ppm for targeted detection. Data processing was performed in Python (v3.10.12) using pyOpenMS (v3.1.0), pyteomics (v4.7.5).

### Protein purification

Transformed *E. coli* strains were grown in LB media with 50 μg/mL kanamycin overnight at 37 °C. Five milliliters of culture were seeded into 500 mL of LB and grown to an $OD_{600}$ of ~0.5. For *spiR*, expression was induced with 400 μM IPTG, whereas *ismA* expression was induced with 100 μM IPTG supplemented with 100 μM l-arginine at the time of induction. Both proteins were incubated overnight at 25 °C. The cells were pelleted for 20 min at $3260 \times g$, and the supernatant was discarded. For SpiR, cells were frozen at −20 °C and subsequently thawed on ice, then resuspended in a lysis buffer of 0.1 M sodium phosphate, 0.2 M sodium chloride, 20 mM BME, pH = 6.5, whereas IsmA was resuspended in 20 mM Tris, 0.2 M sodium chloride, 20 mM BME at pH 8.5. Both slurries were treated with lysozyme, benzonase (Sigma-Aldrich, E8263-5KU), and protease inhibitors. Buffer conditions were optimized based on each protein's isoelectric point to ensure stability, and all comparative assays used buffer-matched controls so that observed differences reflect intrinsic enzymatic properties.

The samples were sonicated at 4 °C using a Branson 550 sonicator at 40% amplitude with 15 s on and 15 s off for a total of 2 min. The cell debris was centrifuged at $14,000 \times g$ for 10 min, and the supernatant was filtered through a 0.22 μm PES syringe filter (Celltreat, 229746). The resulting solution was passed through a Nickel HisTrap Excel column (Cytiva, 17371205) attached to a peristaltic pump. Nonspecific binding was eliminated using 50 mM imidazole (Fisher, BP305). The protein was eluted using 0.2 M imidazole. The SpiR eluent was concentrated, and the buffer was switched to 0.1 M sodium phosphate, 0.2 M sodium chloride, and 20 mM BME using Amicon ultra centrifugal filter 30 kDa MWCO (Millipore Sigma, UFC5030). The IsmA eluent was concentrated, and the buffer was switched to 20 mM Tris, 0.2 M sodium chloride, 20 mM BME at a pH of 8.5. Results of the purification are demonstrated in SDS-Page gels (Fig. S1).

### Binding characterization

The protein stocks were concentrated to the desired concentrations in the respective buffers. All the compounds tested with ITC were prepared in the same buffer as the protein, SpiR, or IsmA tested. Most of the tested compounds were hydrophobic and required additional preparation to ensure proper solubility. Hydrophobic compounds were suspended in glass vials in phosphate buffer as 10 mM stocks and shaken for 48 h at RT. The compounds were diluted to the desired concentrations, and all solutions were degassed prior to use.

Binding experiments were performed at 25 °C using a TA Instruments Nano ITC instrument. Inverse titrations were performed for pregnenolone (Selleck Chemicals, S1914) and 5β-dihydroprogesterone (Cayman Chemical, 34644). The ligand was present in the sample cell, whereas the protein was present in the syringe. Protein was delivered

into the cell in 2.5 uL injections, with a spacing interval of 180 s and a syringe speed of 150 RPM. NADH (Cayman Chemical, 16078), NAD⁺, coprostanone (Chemscence, CS-0643978), and cholesterol (AA Blocks, AA00EBSJ) were used as standard titrations, and the ligand was titrated into the protein. Cholestenone traces had a spacing interval of 120 s, but otherwise conserved the same parameters.

To improve the solubility of cholesterol, cholestenone, and coprostanone in aqueous solutions, 5% (v/v) DMSO was added to the buffer used for the ligand and protein. The initial titrations were performed with 20 injections, which led to aggregation. To mitigate this problem, the reaction time was decreased by titrating 3.1 μL injections over a total of 16 injections. The same method was used for NADH and NAD⁺.

Subtraction of the heats of dilution and analysis was performed using NanoAnalyze software (TA Instruments). The measured heat was integrated into a binding curve, wherein the enthalpy change ($\Delta H$), entropic change ($\Delta S$), stoichiometry ($n$), and binding ($K_d$) were calculated based on the fitting of the curve using the independent binding model for all ligand–protein interactions. The respective protein and ligand concentrations used are described in the figure captions.

## Protein gel electrophoresis

Samples from lysate assays, along with a vector control, were collected at various time points for SDS-PAGE analysis to support the product conversion data. Samples were collected at three key stages: pre-induction, post-induction, and post-lysis supernatant, to monitor protein expression. The samples were prepared by mixing 20 μL of the experimental conditions, 8 μL of Lithium Dodecyl Sulfate (LDS) 4X (GenScript, M00676-10), and 4 μL of Sample Reducing Agent (SRA) 10X (Bioworld, 21440022-2). The samples were then incubated at 95 °C for 5 min. Bio-Rad Mini-PROTEAN® TGX™ Precast Protein Gels, 4–15% (Bio-Rad, 4561084) were prepared, and 8 μL of samples were individually loaded into each well, along with a molecular weight marker (Cell Signaling Technology, 59329S) for protein molecular weight determination. The gels were run for 45 min at 120 volts and subsequently stained with 1% Coomassie Blue Stain (Research Products International, B43000-25.0) in a solution containing 65% deionized water, 35% methanol, and 5% acetic acid (LabChem, LC101003). The results for the samples are described in the figure captions.

## Structural prediction and analysis

The structure of SpiR was predicted using AlphaFold2[49]. Binding pockets were predicted using fpocket (v4.0) with the default parameters[50]. The pockets were compared to the homologous DP-D-glycero-D-manno-heptose 4,6-Dehydratase (PDB: 7US5)[51] to identify the putative substrate-binding regions and catalytic residues. The structures for cholesterol (PubChem compound identifier: 5997) and NADH (PubChem compound identifier: 439153) were docked onto the predicted SpiR structure using AutoDock Vina (v4.2). The docking simulation was performed within $10 \times 10 \times 10$ Å cubes centered on the center points of the chosen fpocket substrate-binding pocket, with exhaustiveness set to 32. The docking results were visualized using PyMOL v3.1.4.1[52].

## Phylogenetic reconstruction and distribution of SpiR homologs

All representative genomes from the Genome Taxonomy Database (GTDB) (release r226)[53] were downloaded, and protein sequences for each genome were predicted using Prokka[54] (version 1.14.6). All protein sequences were annotated using InterProScan v5.73[55]. We used DIAMOND (v2.1.12)[56] to search for homologous amino acid sequences of SpiR in all GTDB sequences in the superfamily SSF51735 (NAD(P)-binding Rossmann fold domain) and filtered hits based on a cutoff of 500 bit-score.

We aligned SpiR with other previously biochemically characterized HSDHs[28] using MUSCLE v5[57] with default parameters and trimmed

to remove columns with more than 97% gaps with goalign (v0.3.7)[58] (Supplementary Data 1). The trimmed alignment was used to construct a phylogenetic tree using RAxML-NG v1.2.2[59], with 1000 bootstrap replicates and the substitution model LG[60].

## Cross-cohort identification of cholesterol derivatives via retention time and m/z harmonization

We identified the retention times of cholesterol, cholestenone, and coprostanone using LC-MS data from three untargeted metabolomic datasets generated from human cohorts: PRISM (PR000677), HMP2 (PR000639), and PROTECT (ST002471). These datasets were generated using similar protocols, which allowed us to align the retention time and m/z features across cohorts. Our approach follows a comparative annotation strategy similar to that described by Kenny et al.[16].

In the PRISM dataset, we identified cholesterol and cholestenone based on their retention times, as previously reported by Kenny et al.[16] (7.21 and 7.00 min, respectively) and expected m/z values. A third compound eluted at ~7.50 min was initially labeled as coprostanol based on the annotations reported by *Cell Host & Microbe* (2018). However, detailed inspection of its ionization profile, specifically m/z 385.3465 $[M + H]^+$, 409.3441 $[M+Na]^+$, and 369.3516 $[M + H–H_2O]^+$, strongly supports its identity as a coprostanone. Coprostanol is known to ionize poorly under electrospray ionization (ESI) and has not been observed consistently in these datasets, further supporting this reassignment[61].

We then cross-referenced these features in the HMP2 and PROTECT. In HMP2, ME233441 was annotated as cholesterol with a retention time of 7.35 min, consistent with PRISM. In the PROTECT dataset, ME584140 and ME584114 corresponded to cholesterol and coprostanone, respectively, with retention times of 7.59 and 7.87 min, respectively. Based on the consistent retention behavior and m/z values, we subsequently assigned the retention times of cholestenone and coprostanone in HMP2 and cholestenone in PROTECT. These assignments were guided by exact mass values and the presence of expected adducts ($[M+H]^+$ and $[M+Na]^+$) co-eluted at the same retention time, enabling the confident harmonization of compound identities across all three metabolomics cohorts.

## Metagenomic profiling of *spiR* homologs and *ismA* homologs in the human gut microbiome

We built a reference database using 323 *spiR* homolog genes as previously described (Supplementary Data 2). A reference dataset of 188 *ismA* homologs was also collected by aligning previously identified *ismA* homologs with MAFFT (v7.526) and building a profile HMM using HMMER (v3.4)[62]. Sequences from the superfamily SSF51735 were aligned to the HMM profile using hmmsearch, and hits were filtered based on a 348 bit-score threshold.

We downloaded samples from 400 HMP2[35], 151 PRISM[36], and 90 PROTECT[37], which had paired metagenomic and metabolomic data. The metagenomic reads were first processed by trimming the adapter sequences using TrimGalore (version 0.6.7)[63] with default settings. The reads were then mapped to a human reference (assembly T2T-CHM13v2.0) to identify potential contaminants and removed them using Samtools (v1.16)[64]. We removed samples with less than one million total reads after curation. We aligned the reads to the *spiR* homolog and *ismA* homolog reference databases using bowtie2 (v2.4.1)[65]. The number of reads mapped to each reference was summarized by normalizing the number of reads in the sample and multiplying by one million to obtain counts per million (CPM). A sample was considered to have *spiR* homologs or *ismA* homologs present if the CPM was greater than 1.

To determine species-level presence, we compiled all strains belonging to the species containing *spiR* homolog ($n = 308$) or *ismA* homolog ($n = 180$) genes and constructed a reference database using the complete genomic sequences of these strains (Supplementary

Data 3). Metagenomic reads were mapped to this reference genome using the alignment method described previously. A threshold of CPM ≥18,000 was applied to define species presence.

## Statistics and reproducibility

No statistical method was used to predetermine sample size. No data were excluded from the analyses. The experiments were not randomized. The investigators were not blinded to allocation during experiments and outcome assessment.

## Reporting summary

Further information on research design is available in the Nature Portfolio Reporting Summary linked to this article.

## Data availability

The authors confirm that the data supporting the findings of this study are available within the article and its supplementary materials. All genomic data is available at the GTDB. The sequencing data for *Astrobacillus crystallinus* and *Eubacterium sp.* ATCC 21408 is available under BioSample numbers SAMN49799817 and SAMN50847228. All metagenomic and metabolomics data analyzed in this study are publicly available. Metagenomic sequencing data for the PRISM, HMP2, and PROTECT cohort are available via SRA with BioProject accession numbers PRJNA400072, PRJNA398089, and PRJNA436359, respectively. The corresponding PRISM, HMP2, and PROTECT metabolomics data are available via the Metabolomics Workbench (http://www.metabolomicsworkbench.org/) under project numbers PR000677, PR000639, and PR001596, respectively. All LC-MS.xml and .mzML files supporting the findings of the study are deposited under Figshare (https://doi.org/10.6084/m9.figshare.31390782). Source data are provided with this paper.

## Code availability

Relevant code for this study is publicly available at the following GitHub repository: https://github.com/nlm-irp-jianglab/spiR_bioinfo, deposited to Zenodo under the https://doi.org/10.5281/zenodo.18744752.

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

## Acknowledgements

This study utilized the computational resources of the NIH HPC Biowulf cluster (http://hpc.nih.gov). We would like to thank Dr. Apostolos Gittis and Dr. David Garboczi from the National Institute of Allergy and Infectious Diseases at the National Institutes of Health for their valuable support and technical assistance with isothermal titration calorimetry. A.K.J., K.D. and X.J. are supported by the Division of Intramural Research of the NIH, National Library of Medicine. S.L. is supported by the NIH training grant T32-AI089621. B.H., G.A. and S.L. are supported by R35-GM155208. This research was supported in part by the Intramural Research Program of the National Institutes of Health (NIH). The contributions of the NIH authors are considered works of the United States Government. The findings and conclusions presented in this paper are those of the authors and do not necessarily reflect the views of the NIH or the U.S. Department of Health and Human Services.

## Author contributions

X.J., B.H., S.L., and G.A. contributed to the conceptualization of the study. G.A. and S.L. developed the methodology and performed validation. Formal analysis was conducted by A.K.J. and X.J., while investigation was carried out by G.A., S.L., and A.K.J. Data curation was performed by A.K.J. and X.J. The original draft of the manuscript was written by K.D.T., G.A., A.K.J., S.L., and X.J. with subsequent review and editing by K.D.T., A.Z., M.G., G.A., S.L., A.K.J., X.J., and B.H. Visualization efforts were led by G.A., X.J., and A.K.J. Supervision was provided by Y.L., B.H. and X.J.

## Competing interests

The authors declare no competing interests.
