## [Transparent Peer Review file · Nature Communications]

SpiR is a gut microbial enzyme that drives cholesterol conversion

Corresponding Author: Dr Brantley Hall

Version 0:

Reviewer comments:

Reviewer #3

(Remarks to the Author)

The authors have responded to my early critique well. A number of issues remain.

1. The authors make the case that SpiR and not IsmA analogs are the major cholesterol converting enzymes in the gut microbiota. This is based on (i) higher substrate affinity than IsmA; (ii) enrichment in individuals exhibiting cholesterol conversion; and (iii) the observation that IsmA is almost never expressed in the absence of SpiR. One compelling piece of data missing would be the determination of catalytic efficiency of both enzymes (K_{cat}/K_m). The reviewer understands that the measurement of the activities of these two enzymes is challenging. Without such measurements the authors should state that this is a limitation of the study.

Minor issues

Line 21 abstract; "3-keto-steroids to their 3 β -hydroxylated forms" should be changed to "3-keto-steroids to 3 β -hydroxysteroids".

Line 99 define the tetrahydroprogesterone isomer.

Line 192 insert "for binding" before respectively.

Line 209: tyrosine does not act as a nucleophile but is the general acid/base.

Reviewer #4

(Remarks to the Author)

I've been asked to comment on the revisions performed to Reviewer #1's concerns. Overall, I believe that the authors have reasonably addressed all concerns to satisfaction. My assessments are detailed below:

1. From the response letter, I learned that reviewer was interested by the co-occurrence of spiR and ismA in the human microbiome, as reported by the manuscript, and encouraged the authors to assess the impact of either or both spiR and ismA to the system. The authors responded by conducting an additional analysis, as shown in Fig. S8, of the two genes in their predictive model. As the authors pointed out, and to my point of view very clear and supportive of their conclusion, that spiR functions independently, with or without ismA. The authors have also justified the absence of a co-expression experiment due to a realistic technical challenge. I think this is reasonable, and does not impair the overall value of the work. Collectively, I believe that the added analysis and reasoning have addressed the reviewer's curiosity.

2. The reviewer suggested the authors to add an analysis of spiR vs ismA's binding affinity to coprostanone and coprostanol. The authors directly implemented this suggestion by performing an additional ITC experiment, and the result (Fig. S2)

showed that spiR has higher affinity than ismA. This analysis further strengthened the findings.

3. The reviewer pointed the authors to two papers discussing an alternative cholesterol metabolism pathway in the human microbiome, and suggested the authors to discuss this in the manuscript. The authors responded by added multiple statements to the Discussion section. I think the statements are objective and informative, bringing the reader's attention to frontiers of the scientific research.

4. The reviewer requested clarification on the different buffers used for purifying IsmA and SpiR proteins. The authors explained the choices of buffer based on the biochemical properties of the protein molecules, and the troubleshooting process they experienced during the study. I think the clarification is sound.

5. The authors fixed a minor overlooking noted by the reviewer. Specifically, they included 3beta-HSDH/I in Fig. 1C.

6. The authors adopted the reviewer's suggestion by including pregnenolone in the chromatogram in Fig. 2C.

7. The authors explained the absence of quantitative peaks in Fig. 2 and 3, as this information requires additional calibration and internal control. Otherwise the quantitations can be misleading. This justification is sound to me.

Reviewer #1 (Remarks to the Author):

Summary: In the present work, Levy and Arp et al. identified a novel microbial steroid delta5-4 isomerase/3-keto reductase (*spiR*), characterized its substrate and stereospecificity, and investigated its prevalence in human microbial communities in relation to cholesterol conversion. Importantly, the authors investigated *spiR* in the context of the previously identified cholesterol converting enzyme (*ismA*), reporting that *spiR* could be a more relevant biomarker and mechanism for microbial cholesterol conversion. While the work in its current form is exciting, novel, and significant to the field of microbial cholesterol metabolism and microbial metabolism at large, there are multiple opportunities to strengthen the findings and provide greater context for this pathway in the development and progression of human disease.

Major comments/concerns:

1. Later in the manuscript (Fig. 6, Fig. S8), the authors report that *spiR* and *ismA* frequently co-occur in the human microbiome. This finding motivates studies similar to those undertaken in Fig. 2 and 3 when *spiR* and *ismA* are co-expressed by bacteria to address whether there is an additive or synergistic metabolic effects. This may be especially interesting in the apparent discrepancy between the production of epipregnanolone by *ismA* from 5beta-dihydroxyprogesterone (Fig. 3c). Additionally, the authors could consider including coexpression of *spiR* and *ismA* homologs and containing species in their metagenomic and ROC analyses (Fig 6C-F).

We thank the reviewer for this valuable suggestion. In this study, we are focusing on defining and characterizing *SpiR*, contrasting it with *IsmA* when appropriate to highlight the differences between these enzymes. While we agree that the future studies should investigate the physiological role of *IsmA* and how *SpiR* interacts with other enzymes, our present study provides a necessary first step in establishing *SpiR* as a mechanistically defined and functionally distinct enzyme in microbial cholesterol metabolism. We also believe that we have demonstrated that *SpiR* performs discrete, non-redundant functions that do not require potential activity from *IsmA*. The evidence for this includes the biochemical assays that indicate that *SpiR* alone is sufficient to catalyze the transformation of cholesterol and 3-keto steroids, including the production of epipregnanolone from 5 β -dihydroxyprogesterone (Fig. 3C).

To further address the reviewer's comment, we conducted additional ROC and precision-recall analyses evaluating the predictive performance of species that encode *spiR*, *ismA*, both, or either gene (Fig. S8). These analyses demonstrate that species encoding *spiR* alone achieve the highest classification accuracy for coprostanol production (AUC = 0.95; AP = 0.94). In contrast, combinations such as "*spiR*⁺ and *ismA*⁺" or "*spiR*⁺ or *ismA*⁺" yielded lower or similar AUC and AP values. The inclusion of *ismA* did not improve the performance of *spiR*-based models. These results further support the conclusion that *spiR* functions independently and is the dominant enzymatic marker of cholesterol conversion in the gut microbiome.

“Figure S8: Receiver Operating Characteristic curve (ROC) for prediction of coprostanone detection in stool samples from all three cohorts (HMP2, PRISM, and PROTECT) based on the presence and absence of *spiR* homologs, *ismA* homologs, *spiR*-containing species, or *ismA*-containing species and their combinations with varying CPM thresholds. The ROC curve and AUC values were calculated using the Python package scikit-learn (v1.6.1).”

Additionally, we identified a human-derived cholesterol-reducing bacterium, *Astrobacillus crystallinus* SVS042, which produces coprostanol and encodes *spiR* but lacks *ismA*. This finding provides direct genomic and biochemical evidence that SpiR alone is sufficient to mediate cholesterol reduction.

While co-expression experiments would be an interesting direction for future work, complete *in vitro* reconstitution of cholesterol-to-coprostanol conversion remains technically infeasible. This pathway requires a missing steroid 5 β -reductase step, which has not yet been identified. Without this enzyme, co-expressing *spiR* and *ismA* would not reconstitute the pathway or yield interpretable results about potential synergy.

Together, these results clarify that *spiR* functions independently of *ismA* and likely represents the principal enzymatic driver of cholesterol conversion in the human gut microbiome. We have now clarified these points in the revised Results:

Lines 127-134: “We utilized electrospray ionization (ESI) mass spectrometry to assay cholestenone production by transformed *E. coli* strains expressing SpiR and IsmA. To improve the detection of cholestenone, we derivatized it with Girard's Reagent P, as has been done in previous studies^{20,21} (Fig. 2a). *E. coli* lysates containing an empty vector served as a control and

showed no cholesterol conversion, whereas those expressing SpiR efficiently produced cholestenone (**Fig. 2b**). These results confirm that SpiR functions as an oxidoreductase acting on cholesterol to produce cholestenone. A comparison of cholestenone signals between *E. coli* lysates expressing IsmA and SpiR revealed that SpiR generated a higher cholestenone signal, with a marginally significant difference ($U = 9.0$, $p = 0.05$, Mann-Whitney U test)."

Lines 150-159: "When *E. coli* heterologously expressing SpiR was incubated with either 5 α -dihydroprogesterone or 5 β -dihydroprogesterone, the predominant product detected matched the retention times of the corresponding 3 β -hydroxy isomers (isopregnanolone and epipregnanolone, respectively) (**Fig. 3b,c**). These results demonstrate that SpiR functions as a 3-keto reductase, reducing both 5 α - and 5 β -dihydroprogesterone to 3 β -hydroxy tetrahydroprogesterone with consistent stereochemistry across divergent substrate conformations.

Together, these findings establish SpiR as a multifunctional enzyme capable of catalyzing three key transformations in microbial steroid metabolism: oxidation of the 3 β -hydroxyl group in cholesterol and pregnenolone, Δ^{5-4} double-bond isomerization, and stereospecific reduction of 3-keto groups to 3 β -hydroxy configurations."

Lines 326-338: "We evaluated the predictive power of *ismA* and *spiR* homolog presence in relation to fecal coprostanone production and performed receiver operating characteristic (ROC) curve analysis at both the gene and species (encoder) levels (**Fig. 6f**). The presence of the *spiR* homolog alone yielded the highest classification performance, with an area under the curve (AUC) of 0.95, outperforming the *ismA* homologs (AUC = 0.81). Similarly, classifiers based on *spiR*⁺ species achieved higher performance (AUC = 0.87) than those based on *ismA*⁺ species (AUC = 0.82). These results confirm that *spiR* homologs are a stronger and more consistent predictor of coprostanone production than *ismA* homologs, whether assessed directly at the gene level or inferred from the presence of the encoding species. To test whether the co-occurrence of *ismA* and *spiR* enhances predictive power, we performed a similar ROC curve analysis on combinations of *spiR*⁺ and *ismA*⁺ species (**Fig. S8**). These analyses reveal that *spiR*⁺ alone offers the best predictive performance, and that accounting for *ismA* cooccurrence does not improve predictive power. This reinforces the conclusion that *spiR* homologs play a central role in cholesterol metabolism in the gut microbiome, and that *ismA* likely does not synergize with *spiR* in mediating cholesterol conversion."

Lines 256-263: "To further validate this taxonomic pattern, we characterized two cholesterol-reducing isolates capable of converting cholesterol to coprostanol. The first was a human-derived bacterium, *Astrobacillus crystallinus* SVS042 (accession JBPPLN000000000), isolated from the feces of a healthy adult. This strain reduced cholesterol to coprostanol and was placed by phylogenomic analysis within the *Acutalibacteraceae* genus CAG-177(submitted to *Microbiology Resource Announcement*). Genome sequencing revealed the presence of *spiR* but absence of *ismA*, suggesting that *ismA* is not strictly required for cholesterol conversion in the human gut. The discovery of a human-derived, *spiR*-positive cholesterol reducer expands the known host range of this metabolic capacity beyond the previously characterized swine-and rat-associated isolates ^{10,33}. "

2. In the ITC experiments in Fig. 4, the authors have presented the exciting finding that spiR binds cholesterol with greater affinity than ismA. They then show that spiR also binds to coprostanone. The authors should include an analysis of coprostanone with ismA to show whether spiR binds coprostanone with greater affinity than ismA. Further, the authors might consider including cholestanone and coprostanol in these analyses to comprehensively characterize the differences between ismA and spiR. Similarly, the studies conducted in Fig. S2 with pregnenolone and 5beta-dihydroprogesterone might also be conducted with ismA to further distinguish spiR from ismA.

We thank the reviewer for this helpful suggestion. To further strengthen the comparison between SpiR and IsmA, we performed additional ITC experiments with coprostanone and cholestenone. Specifically, we measured binding of coprostanone and cholestenone to IsmA, as well as binding of cholestenone to SpiR (Fig. S2). Together with the cholesterol and coprostanone binding data for SpiR that was already presented, these results confirm that SpiR and IsmA exhibit distinct binding preferences. SpiR binds cholesterol with high affinity, whereas IsmA binds cholesterol at a lower affinity but both enzymes also bind coprostanone at comparable affinities.

Our focus in this manuscript is on contrasting SpiR and IsmA with respect to cholesterol transformation and coprostanone recognition. While a comprehensive survey of all possible sterol and steroid products (e.g., coprostanol, pregnenolone, 5 β -DHP) will be an important part of future work in this field, it is beyond the current scope of this work, and we have clarified this in the text. We believe the new data directly address the reviewer's main point and further support the conclusions of our manuscript.

We have now clarified these points in the revised Results:

Lines 161-181: “To characterize the substrate specificity of SpiR, we evaluated its binding affinity for the key substrates involved in these transformations: cholesterol, coprostanone, pregnenolone, 5 β -dihydroprogesterone, and cholestenone. Pregnenolone and 5 β -dihydroprogesterone were also tested but included here primarily as comparators to cholesterol and coprostanone. Isothermal titration calorimetry (ITC) experiments revealed a higher binding

affinity for cholesterol ($K_d = 539$ nM) and coprostanone ($K_d = 660$ nM) (**Fig. 4b, c**) than for homologous steroid hormones. Pregnenolone, which differs from cholesterol only in the absence of a hydrocarbon tail at C17, exhibited 6.12-fold weaker binding affinity ($K_d = 3.3$ μ M) (**Fig. S2**). 5β -dihydroprogesterone, which lacks the same hydrocarbon tail present in coprostanone, bound SpiR with a 1.34-fold lower affinity ($K_d = 887$ nM) (**Fig. S2**). This suggests a substrate preference for coprostanone over 5β -dihydroprogesterone, although this preference is less marked than that between cholesterol and pregnenolone. These differences in binding affinity are likely the result of interactions between the enzyme and the functional groups decorating the steroid core, despite the similarities in the core structures. However, SpiR did not detectably bind to cholestenone, consistent with its role in product release following cholesterol oxidation. (**Fig. S2**). These results indicate that cholesterol and coprostanone are the preferred ligands of SpiR, underscoring their likely role in gut microbial cholesterol metabolism.

As IsmA was also found to metabolize cholesterol, we compared its cholesterol-binding affinity to SpiR. IsmA titrated with cholesterol showed 1.68-fold lower binding affinity than that observed between SpiR and cholesterol ($K_d = 908$ nM) (**Fig. 4a**). This comparatively lower affinity suggests that even though IsmA may participate in cholesterol metabolism, SpiR is likely the more specialized enzyme for cholesterol reduction in the gut. IsmA also bound coprostanone ($K_d = 417$ nM) and (**Fig. 4c**) but did not bind to cholestenone (**Fig. S2**).”

3. While it is well appreciated that the manuscript is focused on cholesterol conversion to cholestanone (and subsequent metabolism to coprostanone and coprostanol), there has been description of another cholesterol metabolism pathway in the human microbiome (PMIDs: 35982311, 35982310) in which cholesterol is sulfonated by *B. thetaiotamicron*. The authors should acknowledge this in the discussion and highlight how these two distinct microbial metabolic pathways for cholesterol might shape not only host health, but also community dynamics within the gut microbiome.

We thank the reviewer for this suggestion and have updated the discussion to reflect other cholesterol metabolism performed in the gut, as well as the importance of crosstalk between gut commensal microbes.

We have now clarified these points in the revised Discussion:

Lines 391-399: “In addition to this reductive pathway, recent studies have shown that *Bacteroides thetaiotaomicron* can sulfonate cholesterol via a dedicated sulfotransferase system, producing cholesterol sulfate that influences host immune and metabolic processes^{48,49}. This shows that gut microbes employ multiple strategies to transform cholesterol, each with its own potential impact in host physiology and microbial ecology. Moreover, inter-microbial interactions—such as competition for cholesterol or cross-feeding between organisms with complementary transformations—are likely to further shape the metabolic fate of cholesterol in the gut. Understanding how *spiR*-containing organisms interact with other cholesterol-transforming microbes will be essential for mapping the broader ecological and physiological impact of these pathways.”

4. It is notable that the buffers used for protein purification of IsmA and SpiR were different (i.e., SpiR in a sodium phosphate buffer at pH 6.5 and IsmA in a Tris buffer at pH 8.5). Could the authors provide justification for this and provide data to show that the different buffers do not affect the results presented for experiments conducted with the purified proteins?

User-provided sequence:

```

10      20      30      40      50      60
MGSSHHHHH SSSLVPRGSH MSTCWLQGKT VVVTGASGGM GAGIAATLIK KHGCTVIGVA
70      80      90      100     110     120
RNEKKMLKVF DELGETYAKQ FSYELFDVSS KENWEKFAEE LQEKGVKVDV LINNAGILPK
130     140     150     160     170     180
FKRFRDYSYE EIERAMNINF YSCVYSVKTM LPMLLQSSSTP AIINIDSSAA LMTLAGTSMY
190     200     210     220     230     240
SASKAALKGF TEALRVEFQG KMFVGLVCPG FTKTDIFSGQ GDADMSNGAK VMDMISTDCD
250     260     270     280     290
KMVKMIMFGI EHKTPMQVHG FDAHAMSVFN RLMVPVYGSKL FSSIMRMSNV DIFKEVFSF

```

[Documentation / Reference]

Number of amino acids: 299

Molecular weight: 33046.28
Theoretical pI: 8.18

User-provided sequence:

```

10      20      30      40      50      60
MGSSHHHHH SSSLVPRGSH MAKKTVFLTG GTGMGWAAV QEMIKHPNEI NIKMLARKSP
70      80      90      100     110     120
KNEEKLKGM AKPNVQVWVG DLGDYDSILE GVTGSDYVLH IGGMVSPTAD WKPVRTQKTN
130     140     150     160     170     180
IGAAQNICKA VLAQPNAADI KVCYIGTVAE TGRNYPIHW GRCDPIKIS IYDHYAISKY
190     200     210     220     230     240
VAERTFVESG IKNVVMRQS GILYPNLIK NMDPIHFVPI NGVLEWCTVE DSGRLMCNLV
250     260     270     280     290     300
LEDEKGLGA DFWNHFFNIG SGEQYRISNY EFECLLLGTL GLAGPEKLF D PNFITKNFH
310     320     330     340     350     360
GQFYADGDKL EEFLLHFREN L PVKDYFNRLA DQVEFYFKIP RYLPKNLVAA CAKPFMKKIA
370     380     390     400     410     420
STKDFGLDW VATRNPRLS AYYGTIEDWA KIPAKWEDFE IKKFAKTS D ADEFKLDHG Y
430     440     450     460     470     480
DETKPESELD IEDMKQAAKF RGGECLSETM TKGDMATK LK WKC GYCGAEF EAS PALLLG
490     500     510     520     530     540
GHWCEPCYIP QKQWDYDNIA RTNPFPAQVM YPNHTKEETN VYKFDLDFQI DGVKWDIDKH

```

[Documentation / Reference]

Number of amino acids: 540

Molecular weight: 61620.13
Theoretical pI: 6.04

We thank the reviewer for this comment. The buffer conditions for IsmA and SpiR were chosen based on the biochemical properties of each protein, particularly their isoelectric points (pI). IsmA has a higher pI (~8.2) and is more stable at slightly basic pH, whereas SpiR has a lower pI (~6) and is more stable under mildly acidic conditions. Accordingly, we used Tris buffer at pH 8.5 for IsmA and sodium phosphate buffer at pH 6.5 for SpiR to maintain each protein in a soluble, neutrally charged state, thereby minimizing aggregation or denaturation during purification. We have updated the methods to reflect this reasoning.

We initially attempted to purify IsmA under the same phosphate buffer conditions used for SpiR. However, this led to reduced solubility and lower yields, confirming that alkaline Tris buffer is optimal for IsmA stability. Conversely, SpiR exhibited greater solubility and stability in sodium phosphate buffer, consistent with its predicted pI and stability profile.

Importantly, all comparative binding assays (e.g., ITC) were performed with buffer-matched controls to eliminate artifacts from buffer composition. Thus, the observed biochemical differences between SpiR and IsmA reflect intrinsic enzymatic properties rather than buffer effects.

We have now clarified these points in the revised Methods:

Lines 492-494: “Buffer conditions were optimized based on each protein’s isoelectric point to ensure stability, and all comparative assays used buffer-matched controls so that observed differences reflect intrinsic enzymatic properties.”

Minor comments/concerns:

- In Fig. 1, the authors make the case that spiR shares sequence similarity to other steroid transforming enzymes such as 3beta-HSDH/I, AcmA, Rv1106c, and ismA. It is unclear why 3beta-HSDH/I is excluded from the structural identity matrix in Fig. 1C while it is included in the pairwise sequence identity in Fig. 1B.

We apologize for the oversight and updated Figure 1C to include 3β-HSDH/I.

- In Fig. 2C, could the authors include a chromatogram for the pregnenolone precursor as they did for cholesterol in Fig. 2B?

We thank the reviewer for this suggestion and have included pregnenolone as a standard in the chromatogram for figure 2C.

7. While the chromatograms in Fig. 2 & 3 are useful for visualization, could the authors also provide quantitation of the peaks? Said differently, could the authors plot peak heights or areas to help the reader approximate the relative differences in product production by vector, *isma*, and *spiR*?

We appreciate the reviewer's suggestion to provide quantitative peak integration. The purpose of Figs. 2 and 3 was to illustrate qualitatively distinct product profiles resulting from vector control, *isma*, and *spiR* expression, rather than to provide absolute or comparative yield estimates. Accurate quantitation of steroid products by LC-MS requires calibration to authentic standards and internal controls to correct for ionization efficiency and matrix effects, which vary substantially among sterols. Without such calibration, comparing raw peak areas or heights across different analytes can be misleading.

For this reason, we chose to focus on maintaining identical chromatographic and instrument settings across replicates, performing analyses in biological triplicate, and validating product identity by high-resolution mass spectrometry. Together, these controls ensure that the observed product differences reflect reproducible enzymatic activity rather than technical variation. We have clarified this rationale in the revised manuscript.

Reviewer #2 (Remarks to the Author):

Hall and colleagues present an enjoyable read and a well-written manuscript with foundational information on gut microbial cholesterol metabolism that could serve as fundamental knowledge for continued functional discoveries in the field of diet-microbiome interactions and host cholesterol homeostasis. The paper focuses on the identification and characterization of a steroid delta 5-4 isomerase/3-keto reductase (*SpiR*) that the authors demonstrate can convert

cholesterol to cholestenone using the appropriate chemical biology and biochemistry-based analysis techniques. This characterization allowed authors to establish *spiR* as a better associated marker of gut microbial cholesterol-coprostanol conversion than the previously characterized *ismA*. Strengths of the manuscript are in the foundational information that this characterization provides for future research.

1. Authors mention that culturable *spiR* gut microbes would make for promising probiotic candidates. Can authors point out *spiR* containing microbes separate from candidates that have *ismA* that would fit this criteria?

We appreciate the reviewer's thoughtful comment regarding probiotic potential. We agree that translational applications such as probiotics must await further delineation of the complete cholesterol-to-coprostanol pathway. In the revised manuscript, we have removed the sentence: "*Culturable members of the spiR-positive clade could represent promising candidates for next-generation probiotics designed to enhance cholesterol excretion,*" to avoid overstatement.

Nonetheless, we believe that *spiR*-containing species remain promising leads, as they are strongly associated and consistently associated with microbial cholesterol conversion (Fig. 6b–f), regardless of whether they encode *ismA*. As shown in Figure 6a, there are genera within the Acutalibacteraceae family that encode *spiR* homologs but not *ismA*. For example, genera *RUG714*, *RGIG2774*, *DTU053*, *UBA1081*, *Fimivicinus* includes species with *spiR* but lacking *ismA*. In addition, we have isolated and sequenced a human-derived cholesterol-reducing bacterium, *Astrobacillus crystallinus* SVS042, which encodes *spiR* but lacks *ismA* and is capable of converting cholesterol to coprostanol. This discovery demonstrates that *spiR* alone can support cholesterol reduction and identifies a culturable *spiR*⁺/*ismA*⁻ species that represents a realistic starting point for future probiotic development.

We have now clarified these points in the revised Results and Discussion:

Lines 242-272: "We performed a survey of the *spiR*-like gene family and identified them exclusively in bacteria from the Clostridia class and Oscillospirales order in the Acutalibacteraceae family (**Fig. 6a**). Acutalibacteraceae are prevalent in anoxic gut environments and are linked to bile acid metabolism in the large intestine, where they play a role in deconjugation and transformation pathways²⁰. The Acutalibacteraceae family is primarily represented by uncultured genera, consisting of nearly all genomes derived from metagenomes (96%, 1096 out of 1144) suggesting that much of the diversity within this family and the cholesterol-reducing microbial community has not been explored. Among the identified species, 308 contained *spiR* homologs and 180 possessed *ismA* homologs. Most *spiR* homologs (314/317) were encoded by species in one Acutalibacteraceae clade, and 148 species within

this clade encoded both *ismA* and *spiR* homologs, indicating a significant overlap between the distribution of both gene types (**Fig. 5b**). The restricted taxonomic distribution and frequent co-occurrence of *spiR* homologs and *ismA* homologs suggest that this clade represents a niche-adapted lineage specialized for cholesterol metabolism in the anaerobic gut, consistent with earlier reports describing non-spore-forming, strictly anaerobic *Eubacterium* species that depend on sterols as essential electron acceptors^{10,12}.

To further validate this taxonomic pattern, we characterized two cholesterol-reducing isolates capable of converting cholesterol to coprostanol. The first was a human-derived bacterium, *Astrobacillus crystallinus* SVS042 (accession JBPPLN000000000), isolated from the feces of a healthy adult. This strain reduced cholesterol to coprostanol and was placed by phylogenomic analysis within the *Acutalibacteraceae* genus CAG-177(submitted to *Microbiology Resource Announcement*). Genome sequencing revealed the presence of *spiR* but absence of *ismA*, suggesting that *ismA* is not strictly required for cholesterol conversion in the human gut. The discovery of a human-derived, *spiR*-positive cholesterol reducer expands the known host range of this metabolic capacity beyond the previously characterized swine-and rat-associated isolates^{10,33}. We also sequenced the genome of *Eubacterium* ATCC 21,408 (accession SAMN50847228), a rat-associated cholesterol-reducing bacterium originally described by Eysen and colleagues^{12,33}. Phylogenomic analysis placed this strain within the genus *Fimencus* of the *Acutalibacteraceae* family. Similar to related taxa, *Eubacterium* ATCC 21,408 encodes both *spiR* and *ismA* homologs (Supplementary Table 2), providing an independent example of their co-occurrence within a single lineage. Together, these two sequenced isolates, one human and one rat-associated, corroborate that cholesterol reduction to coprostanol is a conserved metabolic trait within the *Acutalibacteraceae* and reveals the presence of *spiR*-encoded reductases mediating cholesterol conversion across diverse mammalian hosts.”

Lines 413-421: “If such causal links are established, *spiR*-containing species could represent promising candidates for next-generation probiotics designed to enhance cholesterol excretion^{16,50}. Future work should aim to isolate and study these organisms. Moving toward this goal will require isolating and characterizing *spiR*-positive taxa, as well as testing whether dietary interventions can selectively enrich them as a potential prebiotic strategy. For example, plant-derived sterols are metabolized by *Eubacterium* species and have been shown to influence both microbiota composition and host lipid profiles⁵¹. The precise delineation of the microbial genes and lineages responsible for cholesterol-to-coprostanol conversion in the gut lays the groundwork for exploring translational applications, while maintaining appropriate caution until causal roles are firmly established.”

2. More enthusiasm for the work would be provided if authors were able to go beyond previous associations done using *ismA* as a marker of relevant gut microbial cholesterol-coprostanol transformations and show that *SpiR* is mechanistically relevant to host cholesterol homeostasis. This level of investigation would be needed to support statements such as “[these findings] highlight *SpiR* as a mechanistic driver and predictive marker of microbiome-mediated cholesterol homeostasis”. Even in the cohort data, the intriguing *SpiR* presence/absence associations with cholesterol levels are

just a portion of the information needed to determine that SpiR activity is responsible for stool cholestenone concentrations.

We thank the reviewer for highlighting the importance of connecting microbial cholesterol conversion to host lipid homeostasis. We agree that additional cohort level and experimental evidence would be needed to establish SpiR as a significant driver of cholesterol homeostasis. We recognize that while our data demonstrate strong associations, they do not establish causality. Accordingly, we have tempered our discussion to reflect that *SpiR* is a mechanistic candidate and predictive biomarker, rather than a proven causal driver, while emphasizing that our results lay the groundwork for future causal validation in gnotobiotic and interventional models.

We have modified the discussion accordingly:

Lines 90-91: “These findings redefine the enzymatic basis of microbial cholesterol metabolism and highlight SpiR as a promising mechanistic candidate and predictive marker of microbiome-mediated cholesterol conversion.”

Lines 422-431: “Identification of spiR provides a critical entry point for linking microbial cholesterol metabolism to host physiology. The genes characterized in this study provide a likely route for the microbiome to influence host cholesterol levels, but further work is needed before any robust conclusions can be made. Systemic cholesterol levels are strongly influenced by diet, bile acid metabolism, hepatic synthesis, and host genetics, and further work is required to disentangle the specific contribution of microbial pathways. Translational applications such as probiotics must therefore await further delineation of the complete pathway and further characterization the impact of SpiR on cholesterol conversion. Future studies using gnotobiotic animal models, dietary interventions, and cultivation or synthetic biology approaches to reconstruct the complete coprostanol pathway will be critical for determining whether spiR activity exerts a causal effect on serum cholesterol levels.”

3. There are some strong statements in the paper like - "SpiR-driven variation in coprostanol production between individuals likely contributes to the observed differences in systemic cholesterol levels" that would need much more investigation to warrant including in the manuscript - given that there are many microbiome-independent influences on host cholesterol levels.

We thank the reviewer for this helpful comment. We agree that the original statement overstated the current evidence by implying a direct causal contribution of SpiR activity to systemic cholesterol levels. While our data provide a starting point for understanding these relationships, we recognize that systemic cholesterol levels are determined by numerous microbiome-independent host processes, including dietary intake, bile acid metabolism, hepatic synthesis, and genetic factors.

In response, we have revised the text to soften this statement. Rather than suggesting a direct contribution to systemic cholesterol levels, we now state that SpiR is a mechanistic entry point

into gut microbial cholesterol metabolism and a predictive biomarker of cholesterol conversion potential. We also note that linking SpiR activity causally to systemic cholesterol homeostasis will require additional investigation, such as controlled dietary interventions or gnotobiotic animal studies, designed to isolate microbial effects from host and environmental factors.

We have now clarified these points in the revised Discussion:

Lines 375-378: “As the activating enzyme in the cholesterol reduction pathway, SpiR catalyses the obligate initiating oxidation of cholesterol to cholestenone, forming the electron-withdrawing group required for the next step, the ene-reduction carried out by a 5 β -reductase.”

Lines 400-407: “Systemic cholesterol levels are strongly influenced by diet, bile acid metabolism, hepatic synthesis, and host genetics, and further work is required to disentangle the specific contribution of microbial pathways. Translational applications such as probiotics must therefore await further delineation of the complete pathway and further characterization the impact of SpiR on cholesterol conversion. Future studies using gnotobiotic animal models, dietary interventions, and cultivation or synthetic biology approaches to reconstruct the complete coprostanol pathway will be critical for determining whether spiR activity exerts a causal effect on serum cholesterol levels.”

Reviewer #3 (Remarks to the Author):

This manuscript describes the expression and characterization of two gut microbial enzymes that act as 3 β -hydroxysteroid dehydrogenase/ketosteroid isomerases on cholesterol on route to coprostanol. The study is significant in that coprostanol is a non-absorbable cholesterol metabolite so that expression of these enzymes (SpiR and ismA) could play a role in reducing serum cholesterol levels using the microbiota. The enzyme characterization is reasonably solid, but a number of deficiencies reduce enthusiasm for the work.

1. The authors show that the preferred enzyme SpiR converts cholesterol to cholest-4-ene-3-one. However, the entire pathway to coprostanol involves inversion of configuration at the A/B cis ring junction by an unidentified steroid 5 β -reductase and then inference that the SpiR enzyme converts coprostanone to coprostanol. Without knowing whether the bacteria expressing SpiR also express the 5 β -reductase the introduction of probiotics to enrich for these enzymes becomes problematic.

We thank the reviewer for this important comment. We fully agree that coprostanol formation requires additional enzymatic steps beyond the initial SpiR-catalyzed oxidation, including the action of an unidentified steroid 5 β -reductase. Our study does not claim that SpiR alone completes the pathway. Rather, we view the conversion of cholesterol to cholest-4-en-3-one as the critical activating step in this process, because it introduces the 3-keto group that withdraws electron density and thereby enables subsequent ene-reduction and stereochemical inversion at

the A/B ring junction. In the anaerobic gut environment, this reaction further establishes cholesterol as a terminal electron acceptor, anchoring the pathway as a net electron-gaining process that supports microbial metabolism.

In addition, we showed that *spiR* is a clade-specific gene within Acutalibacteraceae, and speculated that species in this clade are likely to encode the missing 5 β -reductase. This restricted phylogenetic distribution provides a clear genomic context and narrows the search space for identifying the enzyme(s) responsible for the cholestenone-to-coprostanone conversion. Thus, while the precise identity of the 5 β -reductase remains unresolved, our results provide both a functional anchor point (SpiR as the gatekeeper enzyme) and a defined phylogenetic context that will accelerate future discovery of the downstream reductases.

We have revised the Discussion:

Lines 375-385: “ As the activating enzyme in the cholesterol reduction pathway, SpiR catalyses the obligate initiating oxidation of cholesterol to cholestenone, forming the electron-withdrawing group required for the next step, the ene-reduction carried out by a 5 β -reductase. Together, this initial oxidation and the subsequent reductions enable the overall disposal of two electrons to cholesterol. In the anaerobic gut, where electron acceptors are scarce, the utilization of cholesterol as a terminal electron acceptor could be highly advantageous^{33,42,43}. Our findings revealed that this capacity is likely phylogenetically restricted to a clade within the Acutalibacteraceae family, suggesting that it represents a rare trait driving niche adaptation in the gut. This clade therefore represents an ideal target for identifying other lineage-specific genes involved in cholesterol metabolism, including the yet-unidentified 5 β -reductase and potential cholesterol transport systems that enable substrate uptake.”

2. In the purification of the SpiR enzyme the authors show that E.coli extracts catalyze the formation of cholestenone and tetrahydroprogestins. This is not the same as showing that the purified enzyme can perform these reactions. Why were these reactions not performed with the purified homogeneous enzyme?

We thank the reviewer for raising this important point. We agree that demonstrating catalytic activity with purified enzymes would be ideal, and we made extensive efforts to assay purified SpiR under a range of optimized conditions. These included optimization of buffer composition, detergent and cyclodextrin supplementation to improve sterol solubility, cofactor supplementation (NAD⁺, NADH, NADP⁺), and substrate delivery with DMSO, liposomes, and micelles. Despite these attempts, the purified enzyme displayed both low yield and low specific activity, such that reaction volumes required for LC–MS detection of the product would have been impractically large.

A likely explanation is that SpiR is not fully stable in isolation. Sequence and structural analysis revealed a conserved Cys₄ motif consistent with a rubredoxin-like Fe(Cys)₄ center. Such centers are redox-active and prone to oxidation of their Fe(II) state under aerobic conditions, which can disrupt catalytic turnover. Moreover, maintaining correct metal occupancy and redox state often requires the presence of electron transfer partners such as ferredoxins or

flavodoxins. Together, these features help explain why activity was observed robustly in cell extracts, which provide a supportive environment, but not under standard purified conditions.

To overcome these limitations, we assayed activity in *E. coli* lysates, which provided sufficient enzyme abundance and a physiologically compatible environment for sterol solubilization and catalysis. Importantly, the purified protein was used for biophysical assays, including isothermal titration calorimetry with cholesterol/coprostanone, NADH/NAD⁺ binding, biological activity assays using NADH and NAD, and oligomerization analysis, all of which confirmed substrate and cofactor binding consistent with an active enzyme. Together, these results provide strong evidence that SpiR itself is responsible for the observed sterol transformations.

3. The authors state “these results confirm that SpiR functions as a cholesterol oxidase”. Cholesterol oxidase is a FAD dependent enzyme that consumes molecular oxygen and generates hydrogen peroxide. Is this a misstatement?

We apologize for the misstatement. The claim has been updated to reflect the function of the enzyme more accurately. It is now, “These results confirm that SpiR functions as an oxidoreductase acting on cholesterol to produce cholestenone.”

4. SpiR appears to be a member of the SDR family, with a Rossmann fold, and conserved catalytic triad which is well described in the literature. So, the characteristics of the enzyme are not unexpected and thus diminishes the importance of the manuscript with regards to enzyme characterization.

We appreciate the reviewer’s observation. It is correct that SpiR adopts a Rossmann-fold architecture and contains the canonical Tyr-Lys-Ser/Thr catalytic triad typical of the SDR superfamily. However, in response to this comment we performed additional structural analyses, which we have now incorporated into the revised manuscript. These analyses demonstrate that SpiR possesses unique structural features not characteristic of canonical SDR enzymes.

Specifically, SpiR is extended in length relative to most SDR family members and harbors a conserved tetrad of cysteine residues positioned in a geometry consistent with metal coordination. Structural inspection indicates that these cysteines form a pocket highly similar to rubredoxin-like Fe(Cys)₄ centers, which to our knowledge, are unprecedented within the SDR superfamily, whose catalysis generally proceeds without any metal cofactor.

The presence of such a putative redox-active cofactor has important mechanistic and physiological implications. It provides a plausible explanation for the oxygen sensitivity and instability of purified SpiR and suggests that this enzyme is adapted to function in the strictly anaerobic environment of the gut. More broadly, these structural deviations indicate that SpiR represents a functionally specialized, clade-specific SDR variant, expanding the recognized repertoire of catalytic strategies within this ancient superfamily.

We have revised the manuscript to highlight these findings and to emphasize that our contribution is not limited to re-identifying a typical SDR, but to demonstrating that SpiR exhibits novel structural adaptations likely underpinning its central role as the gatekeeper enzyme of microbial cholesterol metabolism.

Lines 213-224: “Unlike canonical SDR enzymes, which are typically 250-250 amino acids in length, SpiR is extended in length with 520 amino acids, and contains an additional C-terminal domain. Within this domain, four cysteine residues (C443, C446, C464, and C467) are conserved across homologs, suggesting strong evolutionary pressure to retain this feature. Structural analysis demonstrated that the cysteines adopt a compact tetrahedral configuration, with sulfur–sulfur separations consistently in the range of ~3.5–3.9 Å. This arrangement is incompatible with [2Fe–2S] or [4Fe–4S] clusters, which require irregular cysteine spacing and bridging sulfides. Although a Zn(Cys)₄ site could, in principle, account for the observed geometry, such motifs typically serve structural rather than catalytic functions, are not universally conserved, and are therefore unlikely in this context. Taken together, the conservation and stereochemistry of these residues are most consistent with coordination of a rubredoxin-like Fe(Cys)₄ center. To our knowledge, this represents a previously unrecognized feature within the otherwise metal-independent SDR superfamily³⁰.”

5. What was the standard enzyme assay used to monitor the purification of the two enzymes? A purification Table should be provided. The authors should provide a SDS-PAGE to show purification to homogeneity.

We thank the reviewer for this reminder and have included SDS-Page gels in the supplemental information (**Fig. S1c and S1d**) that monitor the soluble fraction purification of IsmA and SpiR after being lysed and using a HisTrap column. These gels show clear single bands at the expected molecular weights (~33 kDa for *IsmA* and ~72 kDa for *SpiR*), confirming the purity of the recombinant proteins. We have not included a purification table because we used a straightforward affinity tag–based purification that yields homogeneous protein in one step. As such, there are not meaningful changes in activity, yield, or purity across steps to report. Instead, we have provided direct evidence of homogeneity and activity in the figures and text.

“Figure S1: SpiR and IsmA expression levels monitored by SDS-PAGE band intensity. A broad range of blue pre-stained protein markers (11-250 kDa) was used for protein size determination. (a) Lanes from left to right: IsmA-uninduced, IsmA-induced, IsmA supernatant, SpiR-uninduced, SpiR-induced, and SpiR supernatant. The gel demonstrated a strong band at ~33 kDa in ismA samples and ~72 kDa in SpiR samples that were not present in the vector control (not shown). (b) Lanes from left to right: vector control-induced, vector control supernatant, IsmA-induced, IsmA supernatant, SpiR-induced, and SpiR supernatant. The gel demonstrated an ~33 kDa band for ismA and a ~72 kDa band for SpiR, which were not seen in the vector control lanes. (c) SpiR purification from left to right shows post lysis supernatant, elution fraction 1, 2, 3, 4, 5, 6. This purification was performed on a HisTrap Excel column. (d) IsmA purification from left to right shows post lysis pellet, post lysis supernatant, column flowthrough, elution fraction 1, 2, 3, 4, 5. This purification was performed on a HisTrap Excel column.”

Subsequently, Amicon filters were used to remove smaller protein bands from samples and to concentrate the proteins. 10kD and 30kD filters were used for IsmA and a 30kD filter was used for SpiR.

Minor Issues:

6. Replace tetrahydropregnenolone with a proper name, line 98.

We thank the reviewer for pointing this out and a correction has been made.

7. Use alpha- and betas where appropriate to describe dihydroprogesterone and tetrahydroprogesterone

We thank the reviewer for pointing this out and a correction has been made.

8. Change "flat tetracyclic steroid ring system" to "tetradecahydro-cyclopentaphenanthrene ring system" which is not flat.

We thank the reviewer for pointing this out and a correction has been made.

Reviewer #3 Comments (Round 2):

The authors have responded to my early critique well. A number of issues remain.

1. The authors make the case that SpiR and not IsmA analogs are the major cholesterol converting enzymes in the gut microbiota. This is based on (i) higher substrate affinity than IsmA; (ii) enrichment in individuals exhibiting cholesterol conversion; and (iii) the observation that IsmA is almost never expressed in the absence of SpiR. One compelling piece of data missing would be the determination of catalytic efficiency of both enzymes (K_{cat}/K_m). The reviewer understands that the measurement of the activities of these two enzymes is challenging. Without such measurements the authors should state that this is a limitation of the study.

We appreciate the reviewer's observation and agree this is a limitation in the study. As such, the discussion has been modified to reflect the limitation.

Lines 376-382: "These findings challenge the current *ismA*-centric model of gut microbial cholesterol metabolism and establish SpiR as a key enzymatic contributor to this pathway. Although determination of catalytic efficiency (k_{cat}/K_m) for SpiR and IsmA would provide a more quantitative comparison of their enzymatic contributions, reliable steady-state turnover measurements of these enzymes were not feasible in this study due to technical limitations. We acknowledge this as a limitation of the present study. Even so, the substrate affinity measurements, multi-cohort enrichment analyses, and LC-MS data consistently supports SpiR as the key gut microbial enzyme catalysing cholesterol conversion."

Minor issues

Line 21 abstract; "3-keto-steroids to their 3 β -hydroxylated forms" should be changed to "3-keto-steroids to 3 β -hydroxysteroids".

We thank the reviewer for pointing this out and a correction has been made.

Line 99 define the tetrahydroprogesterone isomer.

We thank the reviewer for pointing this out and a correction has been made.

Line 192 insert "for binding" before respectively.

We thank the reviewer for pointing this out and a correction has been made.

Line 209: tyrosine does not as a nucleophile but is the general acid/base.
We thank the reviewer for pointing this out and a correction has been made.

Lines 208-210: “These residues correspond to the canonical catalytic triad found in the SDR superfamily, in which tyrosine typically acts as a general acid/base catalyst, lysine modulates pKa through hydrogen bonding, and threonine as a proton donor^{2a}.”